# Vision-Zero: Scalable VLM Self-Evolution via Multi-Agent Self-Play

**Qinsi Wang[1], Bo Liu[2], Tianyi Zhou[3], Jing Shi[4], Yueqian Lin[1], Yiran Chen[1],**
**Hai Helen Li[1], Kun Wan[4]\*, Wentian Zhao[4]\***

[1]Duke University  [2]National University of Singapore  [3]University of Maryland  [4]Adobe Inc.
https://github.com/wangqinsi1/Vision-Zero

## ABSTRACT

Although reinforcement learning (RL) has emerged as a promising approach for improving vision-language models (VLMs) and multimodal large language models (MLLMs), current methods rely heavily on manually curated datasets and costly human verification, which limits scalable self-improvement in multimodal systems. To address this challenge, we propose **Vision-Zero**, *a label-free, domain-agnostic multi-agent self-play framework for self-evolving VLMs through competitive visual games generated from arbitrary image inputs.* Specifically, Vision-Zero encompasses three main attributes: (1) **Strategic Self-Play Framework:** Vision-Zero trains VLMs in "Who Is the Spy"-style games, where the models engage in strategic reasoning and actions across multiple roles. Through interactive gameplay, models autonomously generate their training data without human annotation. (2) **Gameplay from Arbitrary Images:** Unlike existing gamified frameworks, Vision-Zero can generate games from arbitrary images, thereby enhancing the model's reasoning ability across diverse domains and showing strong generalization to different tasks. We demonstrate this versatility using three distinct types of image datasets: CLEVR-based synthetic scenes, charts, and real-world images. (3) **Sustainable Performance Gain:** We introduce Iterative Self-Play Policy Optimization (Iterative-SPO), a novel training algorithm that alternates between Self-Play and reinforcement learning with verifiable rewards (RLVR), mitigating the performance plateau often seen in self-play-only training and achieving sustained long-term improvements. Despite using label-free data, Vision-Zero achieves state-of-the-art performance on reasoning, chart question answering, and vision-centric understanding tasks, surpassing other annotation-based methods. Models and code have been released at here.

## 1 INTRODUCTION

Recent breakthroughs in vision-language models (VLMs) have demonstrated remarkable capabilities across diverse multimodal tasks (Achiam et al., 2023; Team et al., 2023). However, current training paradigms face fundamental scalability constraints: they depend heavily on human-curated data through supervised fine-tuning (SFT) (Liu et al., 2023), reinforcement learning from human feedback (RLHF) (Ouyang et al., 2022; Sun et al., 2023), and carefully engineered reward functions for reinforcement learning with verifiable rewards (RLVR) (Guo et al., 2025). This dependency creates two critical bottlenecks. First, a **data scarcity problem**—the extraordinary cost of multimodal annotation limits both scale and diversity of training data, with datasets like COCO Attributes requiring $60,480 for 200,000 objects (Patterson & Hays, 2016), Ego4D consuming over 250,000 annotation hours (Grauman et al., 2022), and Visual Genome mobilizing 33,000 annotators (Krishna et al., 2017). Second, a **knowledge ceiling**—model capabilities remain fundamentally bounded by human-generated supervision, preventing VLMs from discovering strategies beyond human expertise.

**Self-Play** offers a solution by eliminating human supervision through competitive dynamics (Silver et al., 2017; Tesauro, 1995). In self-play, models learn by engaging in competitive interactions with

---

*Corresponding authors.

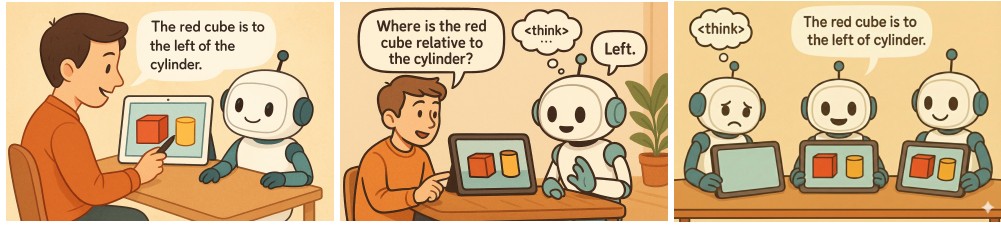

(a) Supervised Learning   (b) Reinforcement Learning   (c) Vision-Zero

Figure 1: **Vision-Zero Paradigm.** (a) Supervised learning depends on human-curated reasoning trajectories; (b) Reinforcement Learning, although enabling models to autonomously learn reasoning processes via validated rewards, still relies heavily on expert-designed question-answer pairs. (c) In contrast, Vision-Zero is a novel self-improvement paradigm entirely independent of human experience. It constructs self-play games by leveraging image pairs that exhibit visual differences. Through the interactive and strategic game, Vision-Zero continuously generates training data for VLMs, enabling the model to achieve scalable self-improvement.

copies of themselves, receiving automatic feedback based on the outcomes of each interaction. As the model improves, its opponents correspondingly advance, thus maintaining a consistently challenging learning environment and driving continuous improvement. By removing the need for human supervision during data generation, self-play has already surpassed the knowledge ceiling across many domains: from TD-Gammon's backgammon supremacy (Tesauro, 1995) to AlphaGo's conquest of Go (Silver et al., 2016; 2017) to OpenAI Five's mastery of complex team coordination (Berner et al., 2019). With the growing capabilities of LLMs, recent work has begun to import Self-Play into LLMs training to reduce dependence on human intervention. These approaches construct **Language Gamification** frameworks wherein LLMs compete under clearly defined game rules, incrementally enhancing their competencies. For example, SPIRAL enhances LLM reasoning by having models play games such as Tic-Tac-Toe and Kuhn Poker (Liu et al., 2025); Absolute Zero frames self-play between proposer and solver (Zhao et al., 2025), achieving state-of-the-art results on mathematics and coding tasks. However, **extending self-play to VLMs remains largely unexplored, despite the prohibitive costs of multimodal data that make such an approach particularly urgent.**

An ideal self-play game environment should satisfy the following four conditions: (1) The skills acquired by agents in order to win the game should closely align with those required by the target tasks. (2) Skill growth should be scalable: as self-play progresses, the environment should continually escalate difficulty so that ever stronger agents can emerge rather than converging to a fixed upper bound. (3) The environment should be sufficiently diverse and complex to enable a wide range of target tasks can satisfy conditions (1). (4) The environment should require no external data or only a small amount of low-cost data, such as label-free data. To the best of our knowledge, existing visual reasoning games fail to satisfy all of the above criteria simultaneously. For instance, Sudoku satisfies conditions (2) and (4), but fails to meet (1) and (3). Due to the multimodal nature of VLMs, designing a self-play environment that fulfills all four conditions requires joint consideration of both vision and language modalities, which is non-trivial. Inspired by language-based social deduction games, particularly those involving alternating rounds of statements and voting such as "Who Is the Spy?", we propose a novel visual reasoning game that addresses these four requirements.

We present **Vision-Zero, the first gamified self-play framework that enables scalable self-improvement of VLMs without requiring human annotations.** We formulate a visual "Who Is the Spy?" game in which the civilians observe an image while the spy receives a blank visual input. To succeed, the spy must infer the hidden visual content solely from the civilians' utterances, while the civilian must balance informativeness and concealment when providing clues grounded in the image. Through this asymmetric interaction, agents progressively develop stronger visual understanding, inference, and communication abilities. This setup compels models to engage in strategic reasoning across multiple roles while handling diverse visual inputs such as CLEVR scenes (Johnson et al., 2017), charts, and natural images. We further propose Iterative Self-Play Policy Optimization (Iterative-SPO), which alternates between Self-Play and RLVR. By incorporating verifiable supervision into self-play, Iterative-SPO stabilizes training and prevents premature convergence to equilibrium states, thereby ensuring consistent performance gains within the Vision-Zero framework.

Vision-Zero provides a domain-agnostic framework that effectively leverages diverse image inputs, enabling continuous improvement without reliance on task-specific datasets. Through a carefully designed strategic visual gameplay, it strengthens reasoning, spatial understanding, and visual comprehension while reducing shortcut bias from text and negative capability transfer that are

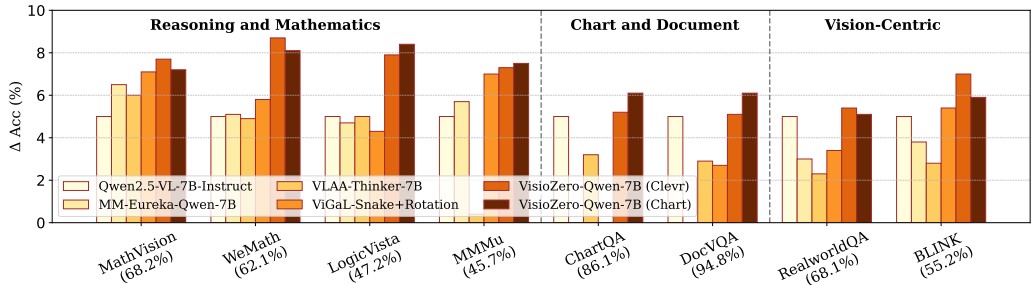

Figure 2: **Performance Comparison of Vision-Zero with SOTA post-training methods.** All models were post-trained on Qwen2.5-VL-7B. The numbers on the horizontal axis represent the accuracy of Qwen2.5-VL-7B on different tasks, while the vertical axis represents the change in accuracy of the trained model. Vision-Zero outperforms baselines trained on expensive human-labeled datasets.

prevalent in conventional VLM training methods. Moreover, Vision-Zero is entirely label-free and eliminates the need for specialized data construction, enabling scalable training directly from raw image inputs. As shown in Fig. 2, Vision-Zero simultaneously enhances performance across tasks including reasoning, chart/OCR, and vision-centric tasks, surpassing state-of-the-art baselines trained on expensive human-labeled datasets. These results underscore Vision-Zero's substantial potential and broad applicability as a pioneering zero-human-in-the-loop training paradigm. Our contributions are as follows:

- We propose **Vision-Zero,** the first gamified self-play framework for VLMs that achieves zero-human-in-the-loop post-training, which supports label-free, domain-agnostic inputs and enables highly cost-efficient dataset construction for scalable optimization.

- We introduce **Iterative-SPO,** a novel algorithm alternating between Self-Play and RLVR to stabilize training and to avoid premature convergence.

- Extensive experiments demonstrate that Vision-Zero substantially enhances model performance across various general tasks, surpassing strong baselines trained on costly human-annotated datasets, especially on reasoning and mathematical tasks.

## 2 VISION-ZERO: A GENERALIZABLE GAMIFICATION TRAINING FRAMEWORK

This section introduces Vision-Zero, a general, scalable, and high-performing gamified VLM post-training framework as illustrated in Fig. 3. We begin by describing the environment and training data (Sect. 2.1). Next, to achieve sustainable performance improvements, we propose Iterative-SPO, which alternates between Self-Play and RLVR (Sect. 2.2). Finally, we provide a comprehensive analysis of the advantages of Vision-Zero compared to human-involved training methods (Sect. 2.3).

### 2.1 ENVIRONMENT AND DATA

**Strategic Environment.** As shown in Fig. 2, Vision-Zero draws inspiration from natural language–based social deduction games, *Who is the Spy*. In this setting, multiple players participate: $n_c$ civilians and a single spy. Each player is assigned an image, where the spy's image is blank. Each round consists two stages:

- **Clue Stage.** In this stage, players are informed of their role (civilian or spy). Each player is then prompted to observe their image and provide a verbal clue that reflects its content such as object descriptions or inferring from the image. Players speak in sequence, and each player's clues become visible to subsequent players; however, their thought processes remain hidden. After multiple rounds clue stage, game enters decision stage.

- **Decision Stage.** In this stage, civilians are instructed to analyze all the provided clues in conjunction with their own image to identify the spy. Since the spy knows their identity, they do not participate in voting. If player is uncertain about who is spy, he can respond with "n/a". Both the reasoning and final votes remain private to players.

*Vision-Zero constitutes a highly strategic and challenging gaming environment.* In the clue stage, the spy must infer the hidden visual content solely from the civilians' clues despite observing only a blank image, and generate plausible clues that are consistent with the shared context to avoid being identified. Civilians must provide accurate, clear clues to avoid suspicion while minimizing information leakage to the spy. During the decision stage, civilians further analyze images and clues meticulously to detect inconsistencies and accurately identify the spy. Detailed prompts for both stages are provided in the Appendix A.2.1 for reference.

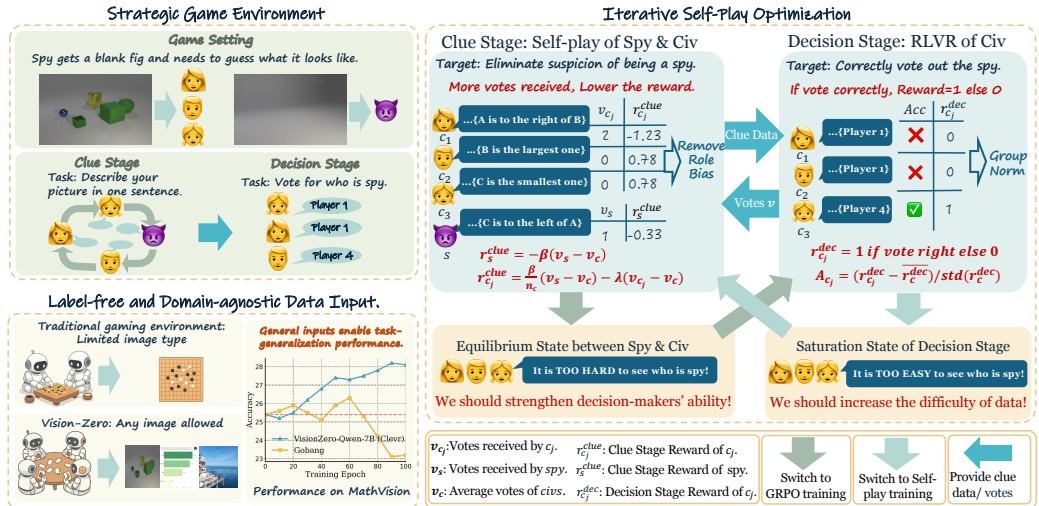

Figure 3: **Overall Framework of Vision-Zero.** Vision-Zero comprises three core components. **Strategic Game Environment:** Each role is required to exhibit strategic behavior tailored to diverse scenarios, thereby simultaneously necessitating multiple capabilities. **Label-free and Domain-agnostic Data Input:** Vision-Zero accepts arbitrary inputs to promote diversity and generalization. To verify this, we train Qwen2.5-VL-7B for 100 iterations on Gobang and our environment and evaluate on MathVision; results show that Vision-Zero effective generalization. **Iterative-SPO:** We introduce a novel two-stage training algorithm. In the clue stage, models are trained via Self-Play using a zero-sum reward inversely proportional to votes received. In the decision stage, models undergo RLVR training with group normalization, using rewards based on vote correctness.

**Label-Free and Domain-Agnostic Data Input.** The input to Vision-Zero is fully label-free yet flexible: for each round, the environment requires only an arbitrary image, where the original image $I_c$ is provided to the civilians, while the spy receives a blank visual input $I_s$. To validate this generality, we experiment with three types of data:

- **CLEVR Data.** (Johnson et al., 2017) We automatically rendered 2,000 images using the CLEVR renderer. Each image contains 4–6 randomly arranged objects. All objects were randomly generated through automated scripting. The entire rendering process required approximately 6 hours on an NVIDIA A100 GPU. Example training set samples are illustrated in Fig. 4 (left).

- **Chart Data.** We randomly selected 1,000 images from the ChartQA (Masry et al., 2022) training set. The dataset includes line charts, pie charts, and bar charts. Examples from this dataset are illustrated in Fig. 4 (middle).

- **Real-World Data.** We randomly sampled 1,000 images from ImgEdit (Ye et al., 2025) training set. Examples from this dataset are shown in Fig. 4 (right).

**Overall, Vision-Zero provides a strategic game-based environment in which the model continuously generates reasoning supervision through interactive gameplay and learns from verifiable rewards, enabling scalable self-improvement. In addition, Vision-Zero supports label-free and domain-agnostic data construction, allowing users to build domain-specific datasets at minimal cost.** As illustrated in the bottom-left of Fig. 3, Vision-Zero achieves sustained performance improvement on the MathVision validation set, outperforming the original model by 3%, which is unattainable in previously narrowly-defined game environments like Gobang.

## 2.2 ITERATIVE SELF-PLAY POLICY OPTIMIZATION

To enable sustained performance improvement within Vision-Zero, we introduce Iterative Self-Play Policy Optimization (Iterative-SPO) which is a novel optimization algorithm that alternates between self-play and RLVR. The workflow of Iterative-SPO is illustrated in Fig. 3.

**Notation.** Assume each round has $n$ players: $n_c$ civilians and one spy, role set is defined as $\mathcal{K} = \{s\} \cup \{c_1, \ldots, c_{n_c}\}$. The spy and civilians hold images $I_s$ and $I_c$, respectively. In clue stage, each player provide clue $u_k \sim \pi_\theta^k(\cdot \mid I_k, h), k \in \mathcal{K}$ based on clue history $h$. In decision stage, a voting mechanism returns vote counts $v = (v_s, v_{c_1}, \ldots, v_{n_c})$, where $v_{c_j}$ represents number of votes $c_j$ received due to being suspected of being spy, and $v_s$ represents the number of votes $spy$ received.

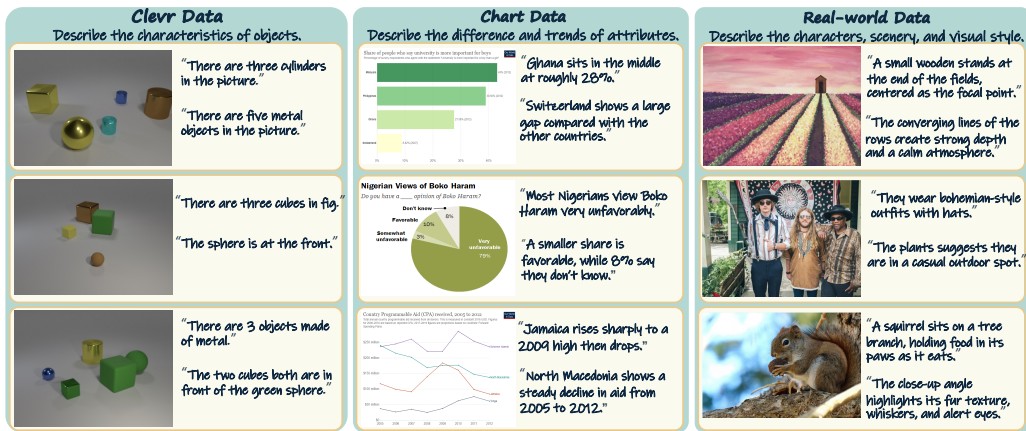

Figure 4: **Visualization of the datasets used in Vision-Zero.** We employ three representative data in our experiments: (left) CLEVR-based data, (middle) Chart-based data, and (right) Real-world data.

**Self-Play Policy Optimization in Clue Stage.** *During this stage, players seek to avoid raising suspicion that they might be the spy.* Moreover, the spy and civilians constitute two opposing sides, and we employ Self-Play Policy Optimization to continuously enhance the model's capabilities.

Zero-Sum Reward. Their rewards are designed according to the zero-sum game principle. Based on these considerations, we define the Clue Stage reward $r_s^{clue}$ and $r_{c_j}^{clue}$ as follows:

$$r_s^{clue} = -\beta \left(v_s - \bar{v}_c\right), r_{c_j}^{clue} = \frac{\beta}{n_c} \left(v_s - \bar{v}_c\right) - \lambda \left(v_{c_j} - \bar{v}_c\right), \qquad j = 1, \ldots, n_c. \quad (1)$$

where $\bar{v}_c = \frac{1}{n_c} \sum_{j=1}^{n_c} v_{c_j}$ denotes the average number of votes received by all civilians, $\beta > 0$ controls the intensity of competition between the spy and the civilians, and $\lambda > 0$ regulates the penalty for behavioral inconsistency among civilians. Eqa.1 ensures that the total reward between the spy and the civilians is zero, and that players receiving more votes are assigned lower rewards.

Role-Advantage Estimation (RAE). To mitigate the imbalance in win probability caused by asymmetric role information, we apply RAE (Liu et al., 2025). Specifically, we initialize RAE coefficient for the spy $b_s$ and the civilians $b_c$ to zero. The RAE coefficient and advantage at each round are:

$$b_s = \alpha b_s + (1 - \alpha) r_s^{clue}, \quad b_c = \alpha b_c + (1 - \alpha) \frac{1}{n_c} \sum_{j=1}^{n_c} r_{c_j}^{clue}, \quad A_k^{clue} = r_k^{clue} - b_k, k \in \mathcal{K} \quad (2)$$

where $\alpha$ denotes the decay rate, and the advantage values $A_k^{clue}$ are computed by subtracting the RAE from the original reward to eliminate information asymmetry.

Objective. With a reference policy $\pi_{\text{ref}}$, the optimization objective of Clue Stage is,

$$\mathcal{L}^{clue}(\theta) = -\mathbb{E}\left[\frac{1}{n} \sum_{k \in \mathcal{K}} A_k^{clue} \log \pi_\theta^k(u_k \mid I_k, h)\right] + \tau_{clue} \mathbb{E}\left[\frac{1}{n} \sum_{k \in \mathcal{K}} D_{\text{KL}}\left(\pi_\theta^k \| \pi_{ref}^k\right)\right]. \quad (3)$$

where the KL term constrains updates to remain close to $\pi_{\text{ref}}$, stabilizing learning and preventing degenerate utterances. Unbaselined returns are zero-sum to promote equilibrium-seeking dynamics.

**RLVR in the Decision Stage.** During this stage, *the objective of each player is to correctly identify and vote for the spy.* Since civilians share aligned information, they can be regarded as a single group. Therefore, we adopt the GRPO objective for Decision Stage.

Discrete Reward. Assume civilians take the full-round clues $H$ and outputs $\hat{s}_{c_i} \sim q_\theta(\cdot \mid H), i = 1, \ldots, n_c$, where $s_{c_i}$ can be index of player (indicating vote for the player as spy), or $\varnothing$ (indicating not clear who is spy and answer "n/a"): assume $s^\star$ is the true spy index. Define reward

$$r_{c_i}^{dec} = +1 \quad \text{if } \hat{s}_{c_i} = s^\star, -0.5 \quad \text{elif } \hat{s}_{c_i} = \varnothing, -1 \quad \text{else.} \quad (4)$$

This reward encourages players to make well-reasoned inferences. Even under highly challenging conditions, it incentivizes acknowledging uncertainty rather than committing to an incorrect answer.

Group Norm & Objective. To remove round-specific difficulty, we apply group normalization:

$$\mu_r = \text{mean}\left[r_{c_i}^{dec}\right], \quad \sigma_r = \text{std}\left[r_{c_i}^{dec}\right], \quad A_{c_i}^{dec} = (r_{c_i}^{dec} - \mu_r)/(\sigma_r + \varepsilon), \quad i = 1, \ldots, n_c \quad (5)$$

where $\varepsilon > 0$ prevents division by zero. With a reference distribution $q_{\text{ref}}$, we optimize the advantage-weighted log-likelihood of the sampled votes with KL regularization:

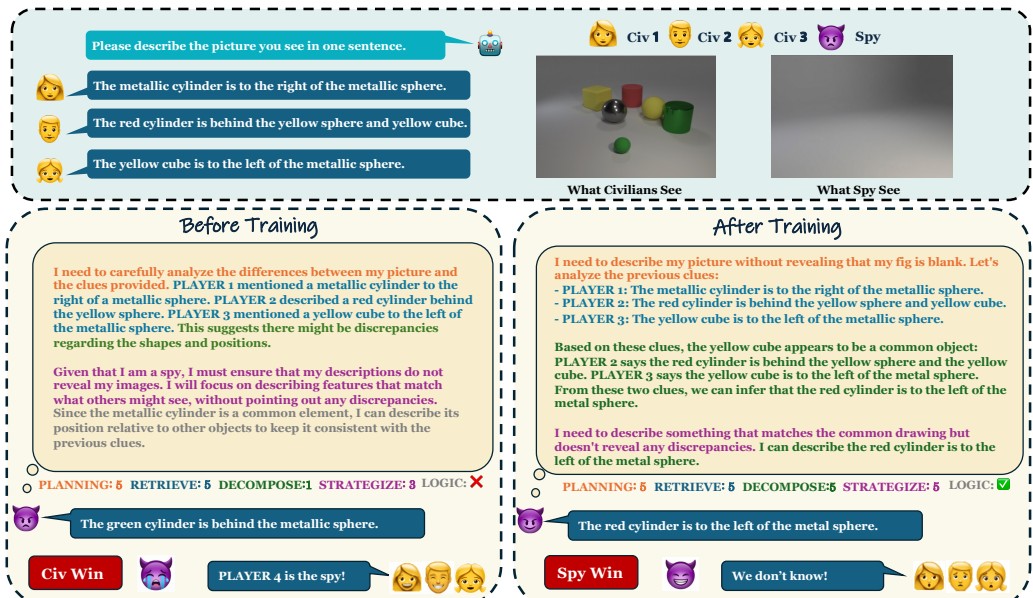

Figure 5: **Visualization of spy reasoning in Vision-Zero.** A comparison of model responses to identical scenarios before and after training, as evaluated by GPT-based scoring, reveals substantial improvements in planning, retrieval, decomposition, strategy formulation, and logical reasoning.

$$\mathcal{L}^{dec}(\theta) = -\mathbb{E}\left[\frac{1}{n_c}\sum_{i=1}^{n_c} A_{c_i}^{dec} \log q_\theta(\hat{s}_{c_i} \mid H)\right] + \tau_{\text{dec}}\,\mathbb{E}\left[\frac{1}{n_c}\sum_{i=1}^{n_c} D_{\text{KL}}(q_\theta(\cdot \mid H)\,\|\,q_{\text{ref}}(\cdot \mid H))\right]. \quad (6)$$

**Iterative Stage Training.** A pure self-play setup typically reaches a local equilibrium (Yao et al., 2023; Balduzzi et al., 2019; Hu et al., 2020; Balduzzi et al., 2018), limiting exploration of new reasoning paths. Conversely, standalone RL methods like RLVR risk knowledge saturation once the available question set is mastered. To mitigate these issues, Iterative-SPO employs a two-stage alternating training. When decision-stage performance indicates clue-stage saturation (easy identification of the spy), training shifts to the clue stage to increase difficulty. Conversely, when identifying the spy becomes challenging, training shifts back decision stage. Let $\mathcal{B}_t = \{(H_i, s_i^\star)\}_{i=1}^B$ be a held-out mini-batch at iteration $t$. Define the average prediction accuracy $\text{acc}_t$ and "n/a" rate $\text{na}_t$ of players in the decision stage within a batch round:

$$\text{acc}_t = \frac{1}{B}\sum_{i=1}^B \mathbf{1}\left[\arg\max_y q_\theta(y \mid H_i) = s_i^\star\right], \quad \text{na}_t = \frac{1}{B}\sum_{i=1}^B q_\theta(\varnothing \mid H_i). \quad (7)$$

We maintain exponential moving averages with smoothing $\rho \in [0, 1)$:

$$\bar{\text{acc}}_t = \rho\,\bar{\text{acc}}_{t-1} + (1-\rho)\,\text{acc}_t, \qquad \bar{\text{na}}_t = \rho\,\bar{\text{na}}_{t-1} + (1-\rho)\,\text{na}_t, \quad (8)$$

initialized as $\bar{\text{acc}}_0 = \bar{\text{na}}_0 = 0$. Let $m_t \in \{0, 1\}$ be the phase indicator ($m_t = 1$ trains the CLUE stage, $m_t = 0$ trains the DECISION stage). We switch phases using hysteresis thresholds $\tau_{\text{acc}}^\uparrow, \tau_{\text{err}}^\uparrow, \tau_{\text{na}}^\uparrow, \tau_{\text{na}}^\downarrow$:

**Decision → Clue:**   if $m_t = 0$ and $\bar{\text{acc}}_t \geq \tau_{\text{acc}}^\uparrow$ and $\bar{\text{na}}_t \leq \tau_{\text{na}}^\downarrow$, then set $m_{t+1} = 1;$   (9)

**Clue → Decision:**   if $m_t = 1$ and $\left(1 - \bar{\text{acc}}_t \geq \tau_{\text{err}}^\uparrow \text{ or } \bar{\text{na}}_t \geq \tau_{\text{na}}^\uparrow\right)$, then set $m_{t+1} = 0;$   (10)

otherwise $m_{t+1} = m_t$. To avoid chattering, we require a minimum dwell time $K_{\min}$ updates per stage. With this gating, the per-iteration training loss is $\mathcal{L}_t = m_t\,\mathcal{L}_{\text{clue}}(\theta) + (1-m_t)\,\mathcal{L}_{\text{dec}}(\theta)$, and gradients are applied only to the active module at iteration $t$. Algorithm is shown in Appendix A.2.3.

This alternating scheme provides two main benefits: (1) It prevents the model from stagnating in a strategic equilibrium or knowledge plateau by dynamically switching training stages upon detecting stagnation signals, thus ensuring continuous improvement (empirically verified in Sect. 3.2). (2) Alternating self-play with RLVR introduces supervised signals, stabilizing training and preventing common pitfalls like role collapse (Wang et al., 2020; Yu et al., 2024) or divergence (Heinrich & Silver, 2016; Vinyals et al., 2019). In summary, **Iterative-SPO provides a stable paradigm that integrates self-play with RLVR optimization to achieve sustained performance improvement.**

## 2.3 ADVANTAGE ANALYSIS

Vision-Zero has three key advantages. Firstly, Vision-Zero leverages **domain-agnostic data inputs** through image differences, allowing it to accept diverse data without reliance on specific image types. This universality enables direct utilization of existing high-quality image datasets, leading

Table 1: **Performance Comparison of Vision-Zero and SOTA models on Reasoning and Math**, evaluated on VLMEvalKit. All results are obtained under same settings, except ViGaL-Snake and ViGaL-Rotation, whose results are obtained from the original paper due to unavailable models. Vision-Zero outperforms baselines trained on extensive manually annotated datasets in related tasks.

| Method | MathVista | MathVision | WeMath | MathVerse | LogicVista | DynaMath | Avg. |
|---|---|---|---|---|---|---|---|
| *Proprietary Model* | | | | | | | |
| GPT4o | 61.4 | 30.4 | 40.0 | 50.2 | 45.9 | 32.3 | 43.4 |
| Gemini2.0-Flash | 73.4 | 41.3 | 57.1 | 54.4 | 56.2 | 43.7 | 54.4 |
| *Performance on Qwen2.5-VL-7B* | | | | | | | |
| Qwen2.5-VL-7B | 68.2 | 25.4 | 36.1 | 49.0 | 47.2 | 20.9 | 41.1 |
| R1-OneVision-7B | 64.1 | 24.1 | 35.8 | 47.1 | 44.5 | 21.4 | 39.5 |
| MM-Eureka-Qwen-7B | **73.0** | 26.9 | 36.2 | 50.3 | 42.9 | **24.2** | 42.9 |
| VLAA-Thinker-7B | 68.0 | 26.4 | 36.0 | 51.7 | 47.2 | 21.9 | 41.9 |
| OpenVLThinker-7B | 70.2 | 25.3 | 36.5 | 47.9 | 44.3 | 21.2 | 40.9 |
| ViGaL-Snake | 70.7 | 26.5 | – | 51.1 | – | – | – |
| ViGaL-Rotation | 71.2 | 26.3 | – | 50.4 | – | – | – |
| ViGaL-Snake+Rotation | 71.9 | 27.5 | 36.9 | 52.4 | 46.5 | 22.9 | 43.0 |
| **VisionZero-Qwen-7B (CLEVR)** | 72.2 | 28.4 | 39.2 | 53.2 | 49.8 | 22.9 | 44.3 |
| **VisionZero-Qwen-7B (Chart)** | 72.4 | 28.9 | 38.8 | 52.5 | 51.2 | 22.1 | 43.3 |
| **VisionZero-Qwen-7B (Real-World)** | 73.1 | 28.5 | 40.1 | 52.1 | 50.8 | 22.5 | 44.5 |

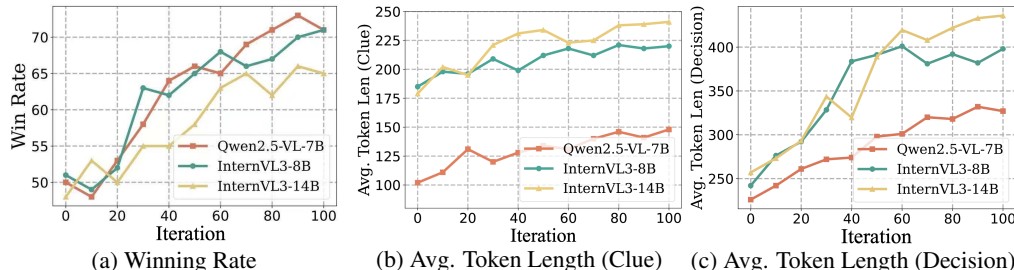

(a) Winning Rate     (b) Avg. Token Length (Clue)     (c) Avg. Token Length (Decision)

Figure 6: **Evolution of win rate and token length during Vision-Zero training.** Win rates are evaluated over 100 rounds (50 civilian, 50 spy) against corresponding untrained reference models; civilians win by correctly identifying the spy. Token length are collected across these rounds.

to generalizable performance improvements at minimal cost, as evidenced by superior benchmark results (Fig. 2). Secondly, Vision-Zero demands **simultaneous analysis of visual and textual inputs**, addressing spatial relationships and object details, thereby concurrently enhancing reasoning, visual comprehension, and OCR capabilities. This integrated approach effectively mitigates common challenges such as text shortcut bias and negative capability transfer, as illustrated in Fig. 5. Lastly, Vision-Zero employs a **highly cost-efficient data curation strategy**, rapidly generating datasets using advanced editing tools like ChatGPT and NanoBanana. This approach significantly reduces costs compared to traditional manual labeling, accelerating practical applications of targeted VLMs.

## 3 EXPERIMENTS

To thoroughly evaluate Vision-Zero, we first outline the experimental setup, the datasets, and the baselines. Next, we evaluate its performance and cost-efficiency across diverse tasks (Sect. 3.1). We then conclude by analyzing model generalizability and the effectiveness of Iterative-SPO. (Sect. 3.2).

**Models, Datasets & Baselines.** We evaluated Vision-Zero using three models—Qwen2.5-VL-7B (Bai et al., 2025), InternVL3-8B, and InternVL3-14B (Zhu et al., 2025)—across 14 tasks in reasoning, chart analysis, and vision-centric domains. Detailed model and dataset information is in the Appendix A.3.1. We compared our models against SOTA methods R1-OneVision-7B (Yang et al., 2025b), MM-Eureka-Qwen-7B (Meng et al., 2025), VLAA-Thinker-7B (Zhou et al., 2025), and OpenVLThinker-7B (Deng et al., 2025) (all post-trained via RLVR on human-labeled data), as well as ViGaL (Xie et al., 2025), which collects game data initially and subsequently training on them.

**Training and Hyperparameter Settings.** We detail the hyperparameters used for Vision-Zero training below. Each round included four civilians ($n_c = 4$) and two clue-stage speeches. To maintain balanced rewards (-1 to 1 range), we set clue hyperparameters $\beta = \lambda = 0.1$. Decay coefficients for role advantage ($\alpha$), accuracy, and "n/a" rates ($\rho$) were adopted from Liu et al. (2025) as $\alpha = \rho = 0.95$. KL regularization weights were set as defaults ($\tau_{\text{dec}} = \tau_{\text{clue}} = 0.04$). Empirically set stage-switching thresholds were $\tau_{\text{acc}}^{\uparrow} = 0.9$, $\tau_{\text{err}}^{\uparrow} = 0.4$, $\tau_{\text{na}}^{\uparrow} = 0.5$, $\tau_{\text{na}}^{\downarrow} = 0.1$, with minimum rounds per stage $K_{\min} = 5$ and patience $P = 20$. Models were trained for 100 iterations with a batch size of 128 using the VLM-R1 (Shen et al., 2025) code framework. Qwen2.5-VL-7B was trained on the CLEVR-based,

Table 2: **Performance comparison between Vision-Zero and other state-of-the-art models on Chart Understanding and Vision-Centric benchmarks.** All models are evaluated using the open-source platform VLMEvalKit. Additional results on related datasets are provided in the Appendix A.4.

| Model | Chart Understanding | | | | Vision-Centric | | | |
|---|---|---|---|---|---|---|---|---|
| | ChartXiV_RQ | FunctionQA | PaperQA | ReachQA | RealWorldQA | MMVP | BLINK | MuirBench |
| *Proprietary Model* | | | | | | | | |
| GPT-4o | 47.1 | 80.7 | 47.4 | 53.3 | 75.4 | 86.3 | 68.0 | 68.0 |
| Gemini2.0-Flash | 61.2 | - | - | 63.0 | 73.2 | 83.0 | 63.5 | 64.6 |
| *Performance on Qwen2.5-VL-7B* | | | | | | | | |
| Qwen2.5-VL-7B | 42.5 | 82.3 | 68.4 | 50.8 | 68.1 | 76.8 | 55.2 | 58.2 |
| R1-OneVision-7B | 35.3 | 69.4 | 64.2 | 46.5 | 58.0 | 61.3 | 48.7 | 46.3 |
| MM-Eureka-Qwen-7B | 43.2 | 79.0 | 73.7 | 51.3 | 66.1 | 74.3 | 54.0 | 61.1 |
| VLAA-Thinker-7B | 41.3 | 79.0 | 68.4 | 50.4 | 65.4 | 71.6 | 53.0 | 57.1 |
| OpenVLThinker-7B | 44.1 | 83.8 | 73.7 | 51.5 | 60.2 | 71.3 | 49.9 | 52.8 |
| ViGaL-Snake+Rotation | 41.8 | 82.3 | 73.7 | 51.8 | 66.5 | 74.6 | 55.6 | 57.8 |
| **VisionZero-Qwen-7B (CLEVR)** | **44.1** | **84.2** | **69.1** | **52.2** | **68.4** | **79.5** | **56.0** | **58.2** |
| **VisionZero-Qwen-7B (Chart)** | **45.8** | **85.5** | **73.7** | **53.4** | **68.5** | **79.1** | **56.8** | **59.2** |
| **VisionZero-Qwen-7B (Real-World)** | **44.8** | **82.3** | **73.7** | **52.8** | **68.9** | **79.2** | **57.2** | **59.4** |

chart-based, and real-world datasets (Fig. 3); InternVL3 was trained only on the CLEVR-based dataset to test generalizability. Further details are provided in Appendix A.3.2.

## 3.1 MAIN RESULTS

**Sustainable Performance Growth.** To verify Vision-Zero's capability to achieve sustained performance growth, we evaluated the models' win rates against a fixed, untrained reference model and measured average token lengths in the Clue and Decision stages on CLEVR data. As shown in Fig. 6, win rates consistently increased during training, with Qwen2.5-VL-7B improving from 50% to 71%. Average token lengths increased substantially, particularly in the Decision stages (e.g., InternVL3-8B and InternVL3-14B grew from 250 to approximately 400 tokens), suggesting enhanced reasoning capabilities facilitated by Iterative-SPO.

**Strong Task Generalization Capability.** To assess whether the performance gains from the Vision-Zero environment generalize to broader reasoning and mathematics tasks, we evaluate our models on six benchmark datasets. The experimental results are presented in Tab. 1. As demonstrated, Vision-Zero models consistently outperform state-of-the-art baseline methods across various benchmarks. Specifically, VisionZero-Qwen-7B (CLEVR) and VisionZero-Qwen-7B (Chart) achieve performance gains of ~3% over the base model, and VisionZero-Qwen-7B (Chart) improves by ~2.2%. In contrast, even the most advanced baseline method yields just ~1.9% improvement. Notably, all baseline methods rely on training with hundreds or even thousands of mathematics and reasoning samples. As a comparison, our Vision-Zero environment does not explicitly include any mathematics-specific task training; rather, it enhances the models' logical reasoning capabilities through strategic gameplay in natural language contexts. These results clearly indicate that the capabilities learned by models from the Vision-Zero environment can effectively generalize to broader mathematics and reasoning tasks, even surpassing models explicitly trained on those large scale task specific datasets.

**Cross-Capability Negative Transfer Mitigation.** A key challenge in VLM post-training is cross-capability negative transfer, where models trained on specific tasks often perform worse on others. As shown in Tab. 2, Vision-Zero-trained models effectively mitigate such negative transfer. Specifically, VisionZero-Qwen-7B (CLEVR) enhances vision-centric task performance, notably increasing MMVP accuracy from 76.8% to 79.5%. Notably, VisionZero-Qwen-7B(Chart) achieves significantly larger gains on chart understanding benchmarks, improving accuracy by an average of +3.9% across the four datasets. VisionZero-Qwen-7B(CLEVR) and VisionZero-Qwen-7B(RealWorld) also exhibit consistent improvements due to enhanced reasoning ability. This demonstrates that Vision-Zero's strategic, multi-capability training environment significantly alleviates negative transfer issues common in traditional single-capability training paradigms. Moreover, thanks to the task-agnostic nature of Vision-Zero, it enables significant performance gains on diverse target tasks such as chart understanding or vision-centric reasoning, through the low-cost construction of task-specific data.

**Low Costs.** Vision-Zero significantly reduces dataset construction costs and training time compared to traditional RLVR methods (Tab. 3). Due to its reliance solely on unlabeled data, Vision-Zero incurs zero labeling costs. In contrast, previous training methods typically require extensive human or model-generated chain-of-thought (CoT) annotations and answer labeling, consuming significant amounts of time and human resources. In addition, due to the high sample efficiency of Vision-Zero, it achieves superior model performance with significantly fewer training iterations. As shown in Tab.

Table 3: **Comparison of dataset construction costs, training costs and model performance across methods.** Label Cost refers to the number of tokens generated by teacher or judging LLMs during data curation; for consistency, all token counts are recalculated using the Qwen2.5 tokenizer. Since VIGAL and Vision-Zero are trained on unlabeled data, they incur no labeling cost. To estimate training time cost, we refer to each baseline's original paper to obtain the number of samples used during RL training, and multiply this by a standard GRPO cost per sample to simulate the expected time consumption under a fully fair setting. For some methods, the value is shown as $\geq$ because we only account for RL cost, excluding SFT. For Vision-Zero, the training time cost is getted by directly measured. Details on all estimates are provided in the Appendix A.3.3.

| Method | Data Cost | | | Training | | | Performance | |
| --- | --- | --- | --- | --- | --- | --- | --- | --- |
| | Prepare Method | Num (RL) | Label Cost (Tokens) | Method | Interact | Time Cost | MMMu | MMMu$_{pro}$ |
| Qwen2.5-VL-7B | – | – | – | – | – | – | 54.3 | 37.0 |
| R1-OneVision-7B | Programmatic construction with human checks. | 10k | $\geq$ 1.1 M | SFT+GRPO | ✗ | $\geq$ 170 A100-Hours | 51.9 | 32.6 |
| VLAA-Thinker-7B | | 25k | 29.6 M | SFT+GRPO | ✗ | $\geq$ 120 A100-Hours | 48.2 | 31.9 |
| OpenVLThinker-7B | | 9k | 5.7 M | SFT+GRPO | ✗ | $\geq$ 125 A100-Hours | 54.8 | 22.1 |
| MM-Eureka-Qwen-7B | | 15k | - | GRPO | ✗ | $\approx$ 700 A100-Hours | 55.8 | 36.9 |
| ViGaL-Snake | Collected in game environment via PPO policy | 72k | 0 | RLOO | ✗ | $\approx$ 170 A100-Hours | 55.8 | 36.6 |
| ViGaL-Rotation | | | | | | | 54.1 | 37.7 |
| ViGaL-Snake+Rotation | | | | | | | 58.0 | 37.4 |
| **VisionZero-Qwen-7B (CLEVR)** | Batch render scenes | **2k** | **0** | **Alternating Self-play+ GRPO** | ✓ | **127 A100-Hours** | **58.8** | **37.7** |

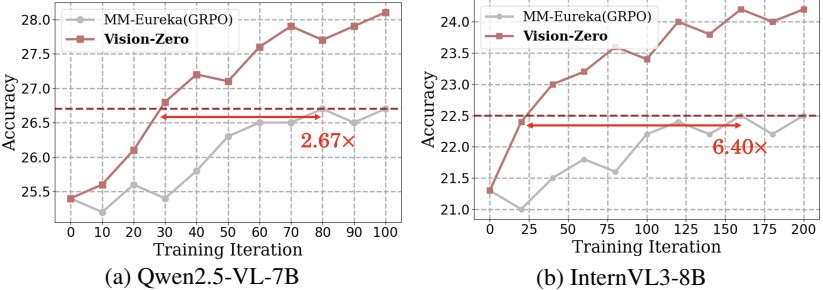

(a) Qwen2.5-VL-7B  (b) InternVL3-8B

Figure 7: **Taining effectiveness comparison between Vision-Zero and the original GRPO.** We compare Vision-Zero and GRPO under identical hardware settings to evaluate training cost and efficiency. Specifically, for the original GRPO, we trained on the MM-Eureka dataset using 8×NVIDIA A100 (80GB) GPUs with a batch size of 128 for 100 iterations on both Qwen2.5-VL-7B and InternVL3-8B. Vision-Zero is trained for the same setting on the Clever dataset using the same hardware. We evaluate the performance of checkpoints from different iterations on MathVista.

3, Vision-Zero requires only 127 A100-hours, which is substantially lower than prior GRPO-based approaches. This underscores Vision-Zero's substantial practical value in real-world applications.

**High Training Efficiency.** Although Vision-Zero involves multi-round interactions, it does not introduce substantial training overhead. On one hand, Vision-Zero adopts a fixed interaction pattern (two clue rounds followed by one decision round), enables fully parallelized forward and backward passes across multiple games, with no asynchronous delays or gradient conflicts. On the other hand, each sample in Vision-Zero generates multiple actions, thereby providing denser learning signals and higher sample efficiency compared to standard single-turn RL setups. To empirically validate the training efficiency of Vision-Zero, we conducted experiments comparing the training time and efficiency of the original GRPO and Vision-Zero over the same number of iterations using identical hardware. As shown in Fig. 7, Vision-Zero achieves markedly higher sample efficiency, resulting in $3.3 \times$ and $6.4 \times$ improvements in overall training efficiency on Qwen2.5-VL-7B and InternVL3-8B, respectively. Furthermore, it yields higher final performance on the validation set. These results highlight the superior training efficiency of Vision-Zero compared to the original GRPO framework.

## 3.2 ABLATION STUDIES

**Model Generalizability.** To assess Vision-Zero's generalizability, we trained InternVL models and evaluated their performance on reasoning and math tasks. Tab. 4 shows VisionZero-InternVL3-8B and VisionZero-InternVL3-14B improved accuracy by 1.8% and 1.6%, respectively, across reasoning

Table 4: **Model generalizability of Vision-Zero.** We train InternVL3-8B and InternVL3-14B within the Vision-Zero using the CLEVR-based dataset. As a baseline, we train InternVL3-8B and InternVL3-14B with vanilla GRPO on the MM-Eureka training set under the same setting as Vision-Zero, and evaluate all models on six reasoning benchmarks.

| Model | MathVista | MathVision | WeMath | MathVerse | LogicVista | DynaMath | Avg. |
|---|---|---|---|---|---|---|---|
| *Performance on InternVL3-8B* | | | | | | | |
| InternVL3-8B | 60.4 | 21.3 | 26.8 | 32.2 | 40.5 | 26.8 | 34.7 |
| MM-Eureka-InternVL-8B | **62.4** | 22.1 | 26.8 | 32.1 | 38.9 | 28.7 | 35.2 |
| **VisionZero-InternVL3-8B** | 62.2 | **24.2** | **28.7** | **32.9** | **41.8** | **29.2** | **36.5** |
| *Performance on InternVL3-14B* | | | | | | | |
| InternVL3-14B | 74.1 | 33.8 | 42.3 | 43.3 | 51.6 | 30.1 | 45.8 |
| MM-Eureka-InternVL-14B | 75.2 | 34.5 | 42.5 | 44.2 | 45.2 | 30.9 | 45.4 |
| **VisionZero-InternVL3-14B** | **75.4** | **34.8** | **44.9** | **45.1** | **53.1** | **31.3** | **47.4** |

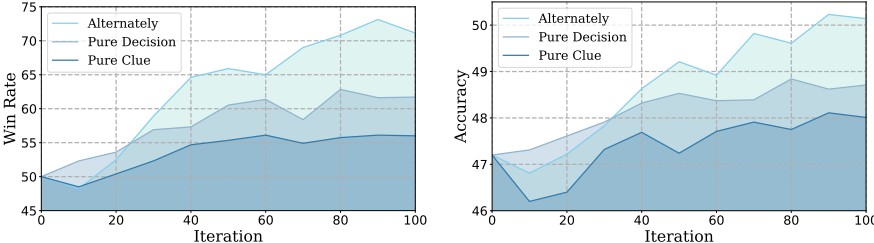

Figure 8: **Performance Comparison between Iterative-SPO and pure Self-play / pure RLVR training.** (left) Winning Rate (right) Performance on LogicVista. We evaluate under three settings: (1) Iterative-SPO; (2) Pure Decision: Clue stage frozen, training only Decision stage via RLVR; (3) Pure Clue: Decision stage frozen, training only Clue stage via Self-Play.

tasks. Compared to the baseline trained using the MM-Eureka dataset and GRPO framework, Vision-Zero consistently enhances the reasoning capabilities of InternVL3-8B and InternVL3-14B models by 1.3% and 2%, respectively. Notably, despite differences in visual encoders, pre-training strategies, and training procedures between the QwenVL and InternVL model series, Vision-Zero consistently improves performance across these models, highlighting its robust generalization capabilities.

**Superiority of Iterative-SPO.** Finally, we evaluate the superiority of Iterative-SPO compared to single-mode training by training Qwen2.5-VL-7B under three distinct settings: (1) Pure clue-stage training: the decision stage is frozen (forward-pass only, without gradient updates); (2) Pure decision-stage training: the clue stage is frozen, with only the decision stage updated; and (3) Iterative-SPO. As shown in Fig. 8, Iterative-SPO substantially outperforms both single-mode approaches, particularly surpassing pure clue-stage training, which experiences slower performance gains and premature equilibrium. This occurs because pure self-play lacks directly verifiable rewards—the reward signal originates from the decision-maker, and when decision quality is insufficient to effectively discriminate roles, the model performance plateaus prematurely. Alternating training mitigates this limitation, achieving sustainable performance improvements; for example, on the LogicVista dataset, it improves final accuracy by 2% over pure self-play and 1% over pure RLVR training.

## 4 CONCLUSION

We introduce Vision-Zero, the first gamified self-play framework for VLMs that achieves zero-human-in-the-loop post-training, addressing self-play training challenges through a strategic environment and domain-agnostic inputs. Our novel Iterative Self-Play Policy Optimization (Iterative-SPO) algorithm alternates self-play with RLVR, incorporating supervisory signals to stabilize training and avoid suboptimal equilibria. Experiments show Vision-Zero significantly improves VLM performance on reasoning, chart/OCR, and vision-centric tasks while substantially reducing dataset construction costs compared to traditional human-labeled datasets, providing an economical, flexible, and robust solution for accelerating VLM development and real-world application.

## REPRODUCIBILITY STATEMENT

We will fully release the model checkpoints and source code to facilitate reproducibility of our results. We provide all prompt design of the gameplay environment in Appendix A.2.1. Implementation details, including the experimental setup, hyperparameters can be found in Appendix A.3. Dataset preparation details can be found in Appendix A.2.2. Algorithm details are shown in Appendix A.2.3.

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

# A APPENDIX

**Organization** In this Appendix, we provide in-depth descriptions of the materials that are not covered in the main paper, and report additional experimental results. The document is organized as follows:

## A.1 RELATED WORK

**Multi-Agent RL for Vision-Language Models.** Self-play has emerged as a powerful paradigm for improving vision-language models without extensive human annotation. Konyushkova et al. (2025) introduce dialog games for VLM self-improvement, where agents engage in goal-oriented play centered on image identification, demonstrating iterative improvement through successful interaction filtering. Foundation model self-play (Dharna et al., 2025) shows how open-ended strategy innovation emerges from competitive interactions between models. SPIRAL (Liu et al., 2025) develops truly online multi-agent multi-turn RL, showing that training on zero-sum games improves reasoning capabilities that generalize to novel downstream tasks—particularly relevant for the multi-turn nature of undercover games. Zhai et al. (2024) present the first framework to fine-tune VLMs using RL with task-specific rewards, achieving state-of-the-art performance without expert data. RL-VLM-F (Wang et al., 2024c) automatically generates reward functions using VLM feedback on image observation pairs, while Rocamonde et al. (2023) demonstrate that VLMs like CLIP can serve as zero-shot reward models with strong scaling effects.

**Undercover and Social Deduction Games in AI.** The undercover game paradigm has been explicitly explored in recent work. Dong et al. (2024) introduce the Multi-Perspective Team Tactic (MPTT) framework for "Who is Undercover?", integrating self-perspective, identity-determination, self-reflection, and multi-round teammate finding to cultivate human-like language expression. Liang et al. (2023) implement an interactive multi-agent framework with human-in-the-loop capabilities, supporting strategic deception and voting mechanics directly applicable to our proposed VLM variant. Studies on social deduction games reveal important insights: Yoo & Kim (2024) demonstrate that GPT-4 achieved 80.65% accuracy in detecting deceivers in Mafia games versus 28.83% for humans, while Kim et al. (2024) identify four major reasoning failures in obscured communication—inadequate information processing, insufficient strategic thinking, lack of theory of mind, and poor temporal reasoning. However, these studies primarily focus on evaluating models' social capabilities through gameplay and rely on prompt engineering to emulate human-like behavior. In contrast, Vision-Zero is the first approach to model the social reasoning game "Who is the Spy" as a self-play environment designed explicitly for training Vision-Language Models (VLMs) to enhance their performance.

**Game-Based Training and Evaluation for VLMs.** Recent benchmarks reveal both the potential and challenges of VLMs in game environments. BALROG (Paglieri et al., 2024) aggregates 6 game environments testing short-term and long-term planning, finding severe deficiencies in vision-based decision-making even for GPT-4o. Wang et al. (2025a) present evaluation frameworks with core tasks directly relevant to undercover game phases: Perceiving, Question Answering, Rule Following, and End-to-End Playing. VideoGameBench (Zhang et al., 2025) reveals frontier models achieve only 0.48% completion rate on popular video games. Novel training approaches leverage games to improve VLM capabilities: G1 (Chen et al., 2025a) introduces VLM-Gym addressing the "knowing-doing" gap through mutual bootstrapping between perception and reasoning during RL training, while JARVIS-VLA (Li et al., 2025) achieves 40% improvement through Act from Visual Language Post-Training.

**Visual Description and Discrimination Tasks.** Description-based discrimination aligns naturally with undercover game mechanics. Menon & Vondrick (2022) introduce "classification by description" using descriptive features rather than broad categories, providing inherent explainability for why agents identify certain images as different. The odd-one-out paradigm directly maps to undercover game structure: Chito et al. (2025) present DINO-based models for spatial and relational reasoning across multiple views, while Mohammadi et al. (2020) develop weakly-supervised tasks showing high correlation with abstract visual reasoning—providing foundations for identifying the different image among a set.

**Multi-Agent Communication in Vision-Language Tasks.** Multi-agent visual communication has seen significant progress. COMMA (Ossowski et al., 2024) presents the first comprehensive benchmark for collaborative work among multimodal agents, featuring vision-language puzzles requiring complementary information access. Qiu et al. (2022) model emergent communication through sketching between neural agents, defining metrics for evaluating conventions applicable to how agents develop shared description strategies. Visual referential games promote systematic generalization: Denamganaï et al. (2023) investigate compositionality with the Obverter architecture, while Lazaridou et al. (2018) show how referential games with pixel input enable linguistic communication emergence, providing theoretical foundations for VLMs learning to describe and discriminate through game play.

**Contrastive RLVR for VLM Post-Training.** Recent work has explored contrastive RLVR as a scalable paradigm for post-training VLMs using automatically constructed image sets. MiCo (Chen et al., 2025b) is a representative example for multi-image reasoning: it constructs image triplets consisting of two augmentations of the same image and a third, similar but different image, prompts the VLM to produce chain-of-thought comparisons and a ternary same/different pattern, and uses an automatically computed accuracy reward under Augmented-GRPO to strengthen fine-grained multi-image comparison. ViCrit (Wang et al., 2025b) instead builds a caption-hallucination proxy task: starting from paragraph-length human captions, it injects a single subtle visual error and trains the VLM, via GRPO and an exact-match span reward, to localize the hallucinated phrase, thereby improving visual perception and hallucination robustness with fully verifiable supervision. GeometryZero (Wang et al., 2025c) introduces group-contrastive policy optimization for geometry reasoning, contrasting auxiliary-construction trajectories within a group to decide when additional constructions are beneficial. Vision-Zero is complementary to these contrastive RLVR approaches. Its gamified setting turns generic unlabeled image pairs into a scalable training environment, beyond the primarily perception- or comparison-oriented gains of prior contrastive RLVR tasks.

## A.2 VISION-ZERO DESIGN DETAILS

Due to space limitations in the main text, this section elaborates on critical implementation details that were previously omitted. We organize the discussion into three parts: (i) the prompt design used during training, (ii) the construction pipeline of training data across domains, and (iii) the formal description of the Iterative Self-Play Policy Optimization (Iterative-SPO) algorithm.

### A.2.1 PROMPT SETTING

The training process simulates gameplay using a sequence of structured prompts, guiding the model through two reasoning stages. We provide below the full prompts used for training on the CLEVR-based visual reasoning task. These prompts are carefully engineered to enforce role-aware generation

and role-consistent reasoning. The design enforces clear behavioral separation between SPY and NOT SPY players, ensuring that the model learns to generate clues consistent with its assigned role and to identify inconsistencies during inference.

For other domains, we adopt the same prompt structure with minimal domain-specific modifications: Chart-based data: The Clue Stage prompt specifies that "two attributes in the chart have been swapped" if the agent is a SPY. This models table-style manipulations in bar plots or pie charts; Real-world image datasets: The SPY instruction is modified to indicate that "some objects or parts of the scene differ" — a softer deviation that reflects the less controllable variation in natural scenes.

In actual training, prompts are dynamically composed. We implement a templating mechanism where placeholder tokens (e.g., player_id, round_num) are automatically filled based on the synthetic episode. Similarly, role-specific insertions (e.g., "You are the spy with the different picture") are programmatically inserted based on the assigned role. This enables batched multi-agent simulation across thousands of games with minimal manual intervention, while still enforcing precise behavioral roles.

---

**CLUE STAGE PROMPT**

You are PLAYER {player_id} of the game "Who is the spy?". This is ROUND {round_num}. Your task is to describe your picture in one sentence, which can only be about the features of objects and the relationship between objects.

Your role: SPY (you have the blank picture)
Your role: NOT SPY (you have the same picture as most others)

**Game Instructions**:
• There are num_players players in total. Only one player sees a different picture, and the pictures seen by all other players are the same.
• Player who are voted as the odd player will be severely punished, so you should convince others that you are not the odd player.
• The odd player (spy) sees a completely blank image with no chart or table at all, while all other players see the same table or chart.

You are the spy (you have the blank picture). Your picture is completely blank (no objects at all), while others see a normal chart or table.

You are not the spy. Your picture is the same as most other players' pictures.

**What you should do:**
1. Compare your picture with all previous clues to understand what others are seeing.
2. If you are NOT the spy: Describe a clear, specific feature of your picture that does not overlap with previous clues. This will help establish what the "normal" picture looks like.
3. If you are the spy:
(1) Infer what topic the chart that others are seeing is about from other players' clues.
(2) Infer what attributes does the chart have and what are the value relationships between these attributes.
(3) Describe a feature that should be correct about the chart that others are seeing but is not same as any previous clue, in order to confuse other players.

**Required output format:**
First, conduct your private step-by-step reasoning — may include suspicions, probabilities, deception, etc. Then, put your one-sentence description of your picture inside \boxed{}. Example answer: \boxed{The cylinder is to the left of the cube}; \boxed{The yellow object is the largest object}; \boxed{The red object is the smallest object}.

---

---

**DECISION STAGE PROMPT**

You are analyzing a game of "Who is the Spy?" where num_players players have just provided clues about their pictures. Your task is to identify which player is the spy (the one with the blank picture) based on their clues.

**Game Setup:**
• There are num_players players in total. Only one player sees a different picture (the spy), and the pictures seen by all other players are the same.
• In the spy's picture, no objects are replaced.
• Each player provided a clue trying to describe their picture while avoiding suspicion.
• The spy tries to blend in by giving clues that seem consistent with what others see.
• Non-spy players try to give distinctive clues that help identify the spy.

**Your Task:**
Analyze all the clues and determine which player is most likely the spy. Look for:
1. Clues that seem inconsistent with the majority
2. Clues that are deliberately vague or generic
3. Clues that might be trying to mislead others
4. Patterns that suggest deception or uncertainty
The image you see shows the ORIGINAL picture (what non-spy players see). Compare the clues against this reference.

**Required Output Format:**
First, conduct your private step-by-step reasoning — may include suspicions, probabilities, deception, etc. Then, put your final answer (PLAYER_NUMBER or N/A) inside \boxed{}. If you are uncertain, you can answer N/A. Example answer: \boxed{1}; \boxed{2}; \boxed{3}; \boxed{N/A}.
All Clues from the Clue-giving Stage:

---

### A.2.2 DATASET PREPARATION

**CLEVR-based Data.** CLEVR is a controlled synthetic environment expressly built to study visual reasoning with minimal dataset bias and rich, program-level supervision. Its images are rendered from complete scene graphs, and the benchmark has become a standard stress-test for multi-step reasoning in vision-language systems (VQA/VLM). CLEVR scenes are procedurally sampled and rendered with Blender in headless mode, emitting both images and a fixed-format JSON that records each object's attributes and pose; the official generator exposes a simple CLI that renders images from the JSON scene specification. This design makes the pipeline lightweight and embarrassingly parallel. The "CLEVR universe" fixes the attribute vocabulary up front. Shapes are from cube, sphere, cylinder. Sizes are small, large. Materials are metal (shiny), rubber (matte). Colors come from an eight-color palette—commonly enumerated as gray, red, blue, green, brown, purple, cyan, yellow—and scenes are populated under simple geometric constraints (no interpenetration, all objects at least partially visible; randomized camera and lighting). These choices simplify perception so models' performance reflects reasoning rather than recognition shortcuts.

We automatically render 2k training pairs with the CLEVR renderer. Each pair consists of an original image and a modified image. Every image is accompanied by its scene JSON; the pair also carries a compact change log (IDs of changed objects and their before/after attributes). For each scene, we sample 4–6 objects with attributes drawn uniformly from the CLEVR spaces above, while enforcing standard CLEVR placement rules (no overlap/interpenetration and sufficient margins so spatial relations are unambiguous). Camera pose and lights are jittered per scene, following the official generator's practice of randomizing viewpoint and illumination. Given an original scene JSON, we randomly select two objects and replace only their color and shape (leaving other attributes and the global layout unchanged unless a minimal nudge is needed to maintain non-overlap). Concretely:

- Step 1: Generate original JSON and render.
- Step 2: Edit the JSON in place for two objects: shape ← new shape, color ← new color.
- Step 3: Re-render with Blender from the modified JSON to obtain the paired image.

CLEVR generation is stateless per scene and the official script supports GPU-accelerated Blender rendering (CUDA flag) in batch mode, so we parallelize across processes. On a single NVIDIA A100, end-to-end rendering of the 2k pairs completes in roughly 6 hours in our environment, consistent with the repo's recommendation to invoke Blender headless with GPU enabled.

**Chart-based Data.** In our preliminary attempts to generate chart data, we explored direct editing of chart images via NanoBanna and ChatGPT; however, we found this approach extremely challenging, because current image editing models and tools struggle to reliably control fine-grained graphical attributes (such as exact axis ticks, bar widths, label alignment, and consistent color scales) without introducing visual artifacts or distortions (a known limitation of current image editing in diffusion/in-painting frameworks) Therefore, to achieve stable, controllable editing and generation, we eventually adopted the following pipeline:

- We let GPT-4o ingest the original chart image and output a JSON file encoding every attribute's numerical value (e.g. data points, axis bounds, legend mapping) as well as auxiliary metadata (chart type, color scheme, layout constraints);
- We prompt GPT-4o to swap two attributes arbitrarily and rewrite the JSON accordingly;
- We feed the new JSON into a Python plotting module to render a new chart.

This paradigm is robust to typical failures of AI editors and fully leverages the strong captioning and scene-parsing abilities of current multimodal LLMs.

For our dataset, we randomly sampled 1,000 original charts from ChartQA's training set to ensure visual and data diversity, so that derived pairs reflect ChartQA's spectrum of chart styles and complexity. ChartQA's dataset spans three canonical chart types — line plots, bar charts, and pie charts — capturing both simple and complex variants in real-world sources. Thanks to the fully automated pipeline, the entire generative process incurs only on the order of tens of US dollars.

### A.2.3 ITERATIVE-SPO ALGORITHM

In this section, we describe the algorithm of Iterative Self-Play Policy Optimization (Iterative-SPO) algorithm, as detailed in Alg. 1. As mentioned in the main paper, Iterative-SPO achieves sustained performance improvement by incorporating supervision signals into the self-play framework through a two-stage alternating training procedure.

---

**Algorithm 1** Iterative Self-Play Policy Optimization(Iterative-SPO)

---

**Input:** Role set $\mathcal{K} = \{\texttt{spy}\} \cup \{c_1, \ldots, c_{n_c}\}$; reference policies $\pi_{\text{ref}}^{\text{spy}}, \pi_{\text{ref}}^{\text{civ}}$; hyperparams $\beta, \lambda, \alpha, \tau_{\text{clue}}, \rho,$
$\quad \tau_{\text{acc}}^{\uparrow}, \tau_{\text{err}}^{\uparrow}, \tau_{\text{na}}^{\uparrow}, \tau_{\text{na}}^{\downarrow}, K_{\min}, P$; learning rates $\eta_\theta, \eta_\theta$.

1: Init RAE $b_s \leftarrow 0$, $b_{\text{civ}} \leftarrow 0$; Stage switch metrics $\bar{\text{acc}} \leftarrow 0$, $\bar{\text{na}} \leftarrow 0$; Stage $m \leftarrow 0$ (Decision).
2: **for** $t = 1, \ldots, T$ **do**
3:     **if** $m = 1$ **then**                                            ▷ **CLUE Stage**
4:         Each player gives clue $u_k \sim \pi_\theta^k(\cdot \mid I_k, h)$ based on the historical dialogue $h$ and input picture $I_k$.
5:         Obtain votes from the decision stage $v = (v_s, v_{c_1}, \ldots, v_{c_{n_c}})$ and $\bar{v}_c \leftarrow \frac{1}{n_c} \sum_{j=1}^{n_c} v_{c_j}$.
6:         *Zero-Sum Rewards:* $r_s^{\text{clue}} \leftarrow -\beta(v_s - \bar{v}_c); \quad r_{c_j}^{\text{clue}} \leftarrow \frac{\beta}{n_c}(v_s - \bar{v}_c) - \lambda(v_{c_j} - \bar{v}_c)$ for $j = 1, \ldots, n_c$.
7:         *Role Advantage Estimation:* $b_s \leftarrow \alpha b_s + (1-\alpha) r_s^{\text{clue}}, \quad b_{\text{civ}} \leftarrow \alpha b_{\text{civ}} + (1-\alpha) \frac{1}{n_c} \sum_j r_{c_j}^{\text{clue}}$.
8:         *RAE-based Advantages:* $A_s^{\text{clue}} \leftarrow r_s^{\text{clue}} - b_s; \quad A_{c_j}^{\text{clue}} \leftarrow r_{c_j}^{\text{clue}} - b_{\text{civ}}$ for $j = 1, \ldots, n_c$.
9:     **else**                                                ▷ **DECISION Stage**
10:         Each citizen casts vote $\hat{s}_{c_i} \sim q_\theta(\cdot \mid H)$ based on the clue information $H$ and the input image $I_k$.
11:         *Reward:* $r_{c_i}^{dec} \leftarrow 1$ if $\hat{s}_{c_i} = s^\star$ (correct); $r_{c_i}^{dec} \leftarrow -0.5$ if $\hat{s}_{c_i} = \varnothing$ (unsure); $r_{c_i}^{dec} \leftarrow -1$ else (wrong).
12:         *Group-norm Advantage:* $A_{c_i}^{dec} = (r_{c_i}^{dec} - \mu_r)/(\sigma_r + \varepsilon)$
13:     *Policy update:* Apply KL-regularized policy gradient as Eq. 3 or Eq. 6 to update $\pi_\theta$ or $q_\theta$.
14:
15:     **Stage Switch:** Calculate average prediction accuracy $\text{acc}_t$ and "n/a" rate $\text{na}_t$ of players in the decision stage within a batch round: $\text{acc}_t = \frac{1}{B} \sum_i \mathbf{1}[\arg\max_y q_\theta(y \mid H_i) = s_i^\star]$, $\text{na}_t = \frac{1}{B} \sum_i q_\theta(\varnothing \mid H_i)$.
16:     Update EMAs $\bar{\text{acc}} \leftarrow \rho \bar{\text{acc}} + (1-\rho) \text{acc}_t; \quad \bar{\text{na}} \leftarrow \rho \bar{\text{na}} + (1-\rho) \text{na}_t, d \leftarrow d+1$.
17:     **if** $m = 0$ **and** $\bar{\text{acc}} \geq \tau_{\text{acc}}^{\uparrow}$ **and** $\bar{\text{na}} \leq \tau_{\text{na}}^{\downarrow}$ **and** $d \geq K_{\min}$ **then** $m \leftarrow 1, d \leftarrow 0$ ;
18:     **if** $m = 1$ **and** $(1 - \bar{\text{acc}} \geq \tau_{\text{err}}^{\uparrow}$ **or** $\bar{\text{na}} \geq \tau_{\text{na}}^{\uparrow})$ **and** $d \geq K_{\min}$ **then** $m \leftarrow 0, d \leftarrow 0$;
19: **return** $\theta, \theta$

---

### A.3 EXPERIMENTS SETTING

In this section, we provide a comprehensive account of the experimental settings used throughout our study. We detail the choices for (1) models, datasets, and baselines, (2) training procedures and hyperparameter configurations.

#### A.3.1 MODEL, DATASET AND BASELINES

**Models.** We evaluate three open-weight vision–language models. Qwen2.5-VL-7B-Instruct is a 7B instruction-tuned VLM from the Qwen family; it upgrades the vision stack with a window-attention ViT and SwiGLU/RMSNorm, and is designed for strong document/chart understanding, structured JSON outputs, grounding, and even long-video/agentic use cases. InternVL3-8B is the 8B member of the InternVL3 series that follows a "ViT-MLP-LLM" design by pairing an InternViT-300M vision encoder with a Qwen2.5-7B language core via an MLP projector; it introduces Variable Visual Position Encoding and native multimodal pre-training to improve multi-image/video perception and OCR/chart/document reasoning. InternVL-14B is an earlier 14B vision-language foundation model (224-px variant) trained on large-scale web corpora such as LAION, COYO, CC12M/CC3M, SBU, and Wukong, and is commonly used for zero-shot classification, retrieval, and captioning baselines.

**Datasets.** We evaluate on a broad suite of public benchmarks. MathVista (Lu et al., 2024) combines 6,141 problems drawn from 28 existing multimodal math datasets plus three newly created sets (IQTest, FunctionQA, PaperQA) to probe fine-grained visual–mathematical reasoning. MathVision (MATH-V) (Wang et al., 2024b) curates 3,040 competition-grade problems with visual contexts across 16 disciplines and five difficulty levels for rigorous multimodal math assessment. We-Math (Qiao et al., 2024) collects 6.5K visual math problems organized over 67 hierarchical knowledge concepts to analyze LMM reasoning behaviors. MathVerse (Zhang et al., 2024b) offers 2,612 diagram-based problems, each converted into six modality variants to stress-test vision vs. text contributions. LogicVista (Xiao et al., 2024) targets logical cognition in visual contexts with 448 multiple-choice questions spanning five task types and nine capabilities, each paired with human rationales. DynaMath (Zou et al., 2025) is a dynamic robustness benchmark that perturbs seed questions (e.g., values, function graphs) to test stability of visual math reasoning. ChartXIV (Wang et al., 2024d) is a realistic chart understanding benchmark comprising 2,323 diverse charts from scientific papers with both descriptive and reasoning questions that stress-test MLLMs beyond template-based chart QA. FunctionQA (Lu et al., 2024) is a visual mathematical reasoning dataset focusing on algebraic reasoning over functional plots, requiring fine-grained interpretation of curves, variables, and equations. PaperQA (Lu et al., 2024) is a scientific reasoning dataset built on academic paper figures, designed to evaluate models' ability to interpret complex visualizations and answer content-based questions about scientific literature. ReachQA (He et al., 2025) is a synthesized chart reasoning dataset containing 3k reasoning-intensive charts and 20k Q&A pairs, constructed to enhance both chart recognition and higher-level visual reasoning in MLLMs. RealWorldQA (Zhang et al., 2024c) (released with Grok-1.5 Vision) contains 700+ real-scene images—many vehicle-captured—each with a question and easily verifiable answer. MMVP (Zhang et al., 2024a) is built from "CLIP-blind" image pairs to assess nine basic visual pattern failures via 300 VQA items. BLINK (Fu et al., 2024) recasts 14 core perception tasks into 3,807 multiple-choice questions that humans solve "within a blink" but remain challenging for current MLLMs. MuirBench (Wang et al., 2024a) focuses on multi-image understanding with 11,264 images and 2,600 MCQs across 12 tasks and 10 relation types, including paired unanswerable variants for robustness.

**Baselines.** We benchmark against five recent multimodal reasoning baselines. R1-OneVision-7B (Yang et al., 2025b) is a Qwen2.5-VL–based VLM trained on the R1-OneVision corpus with a cross-modal reasoning pipeline that converts images into structured textual representations to enable step-wise "R1-style" multimodal reasoning. MM-Eureka-Qwen-7B (Meng et al., 2025) introduces the MMK12 dataset and employs rule-based reinforcement learning with online filtering and a two-stage training strategy to stabilize multimodal math reasoning at the 7B scale. VLAA-Thinker-7B (Zhou et al., 2025) is trained on VLAA-Thinking—a corpus of step-by-step visual reasoning traces with both SFT and RL splits—used to probe SFT vs. RL for R1-like reasoning and reporting SOTA on OpenCompass as of April 2025. OpenVLThinker-7B (Deng et al., 2025) follows an iterative SFT→RL regimen (e.g., GRPO) that consistently improves performance on MathVista/EMMA/HallusionBench, evidencing the synergy of SFT and RL for complex multimodal reasoning. ViGaL (Snake+Rotation) (Xie et al., 2025) post-trains a 7B model purely via RL on

simple arcade-style games (Snake and a 3D rotation puzzle), where combining the two games yields stronger out-of-domain generalization (e.g., math, geometry) than either alone.

### A.3.2 TRAINING AND HYPERPARAMETER SETTINGS

To facilitate stable and effective training, we selected VLM-R1 as the foundational model architecture for the Vision-Zero framework, ensuring compatibility with established benchmarks. The detailed hyperparameter configurations employed in our experiments are summarized in Tab. 5. Specifically, all Vision-Zero models underwent training for 100 iterations across diverse datasets, followed by rigorous evaluation of their post-training performance to measure generalization and robustness.

Table 5: Vision-Zero training hyperparameters.

| Symbol | Meaning | Value |
|---|---|---|
| $n_c$ | Number of civilians per round | 4 |
| $\beta$, $\lambda$ | Clue-stage reward scaling / clue regularization coefficients | 0.1 |
| $\alpha$, $\rho$ | Decay coefficients for role advantage ($\alpha$) and accuracy / "n/a" rates ($\rho$). | 0.95 |
| $\tau_{\text{dec}}$, $\tau_{\text{clue}}$ | KL regularization weights (decoder / clue) | 0.04 |
| $\tau_{\text{acc}}^{\uparrow}$ | Stage-switch (up) threshold for accuracy | 0.9 |
| $\tau_{\text{err}}^{\uparrow}$ | Stage-switch (up) threshold for error rate | 0.4 |
| $\tau_{\text{na}}^{\uparrow}$ | Stage-switch (up) threshold for "n/a" rate | 0.5 |
| $\tau_{\text{na}}^{\downarrow}$ | Stage-switch (down) threshold for "n/a" rate | 0.1 |
| $K_{\min}$ | Minimum number of rounds per stage | 5 |
| $P$ | Patience (number of rounds before forcing change) | 20 |
| # iterations | Total training iterations | 100 |
| Batch size | Training batch size | 128 |

We utilized a training batch size of 128, precisely calculated as the product of `nproc_per_node` (8), `gradient_accumulation_steps` (16), and `num_generations` (8). This carefully chosen batch size aligns with standard VLM training paradigms, effectively ensuring stable optimization dynamics. Moreover, our training setup is fully integrated with state-of-the-art optimization techniques and libraries, including FlashAttention-2 and DeepSpeed ZeRO-3, maximizing training efficiency and scalability while maintaining full methodological consistency with VLM-R1 standards.

```
torchrun --nproc_per_node="8" \
    --nnodes="1" \
    --node_rank="0" \
    --master_addr="127.0.0.1" \
    --master_port="12350" \
    src/open_r1/grpo_jsonl.py \
    --deepspeed local_scripts/zero3_model_parallel.json \
    --output_dir $OUTPUT_BASE_DIR/$RUN_NAME \
    --model_name_or_path Qwen/Qwen2.5-VL-7B-Instruct \
    --dataset_name "dynamic_clevr_spotdiff" \
    --use_dynamic_dataset \
    --epoch_size $EPOCH_SIZE \
    --data_generator_type clevr_spotdiff \
    --clevr_images_dir $CLEVR_IMAGES_DIR \
    --clevr_scenes_dir $CLEVR_SCENES_DIR \
    --clevr_num_players $NUM_PLAYERS \
    --clevr_num_rounds $NUM_ROUNDS \
    --training_phase $TRAINING_PHASE \
    --data_generator_seed 42 \
    --max_anyres_num 6 \
    --max_prompt_length 8000 \
    --max_completion_length 512 \
    --num_generations 8 \
    --per_device_train_batch_size 8 \
    --gradient_accumulation_steps 16 \
```

```
--logging_steps 1 \
--bf16 \
--torch_dtype bfloat16 \
--beta 0.04 \
--report_to wandb \
--gradient_checkpointing true \
--attn_implementation flash_attention_2 \
--num_train_epochs 15 \
--learning_rate 1e-5 \
--warmup_ratio 0.1 \
--lr_scheduler_type cosine \
--run_name $RUN_NAME \
--save_steps 5 \
--save_only_model true \
--reward_funcs clevr_clue_format_with_votes clevr_decision_accuracy \
--dispatch_batches False \
--val_split_ratio 0.0 \
--num_iterations 1
```

### A.3.3 TRAINING COST ESTIMATION OF BASELINES

Here we report how the label cost and training time cost were estimated in Tab.3 of the main text.

**Label cost (tokens).** For label cost, we count tokens generated by teacher or judging LLMs during data curation. For R1-OneVision, VLAA-Thinker, OpenVLThinker and MM-Eureka we directly reuse the token counts reported in Perception-R1 (Xiao et al., 2025). All token counts are recomputed using the Qwen2.5 tokenizer for consistency. For ViGaL, it collects trajectories in game environments and calls GPT-4o only once to draft static reasoning instructions shared across all samples, so we set its label cost to zero. Vision-Zero uses fully CLEVR scenes with rule-based rewards and does not invoke any external LLM during data construction, thus its label cost is zero.

**RL Training time cost (GPU-hours).** All compared methods use GRPO-style RL, so we approximate their RL compute with a unified per-sample cost. From Perception-R1 we know that RL training on $\approx 1.4K$ distinct samples for 25 epochs (about $3.5 \times 10^4$ RL samples in total) consumes 167.4 A100-hours after accounting for judge utilization. This gives an average cost

$$c \approx \frac{167.4}{3.5 \times 10^4} \approx 4.8 \times 10^{-3} \text{ A100-hours per RL sample.} \tag{11}$$

For each baseline, we extract from the original paper the number of distinct RL examples and epochs, compute the total number of RL training samples $N_{\text{RL}}$, and estimate its RL cost as

$$T \approx c \cdot N_{\text{RL}}. \tag{12}$$

When the epoch count is not reported, we conservatively assume one epoch, so the reported GPU-hours are lower bounds. For VisionZero-Qwen-7B (CLEVR), instead of using the above approximation we directly measure the wall-clock RL training time on $8\times$A100-80G GPUs.

### A.4 SUPPLEMENTARY EXPERIMENTAL RESULTS

### A.4.1 COMPREHENSIVE EVALUATION ON CHART/OCR TASKS

While we partially presented Vision-Zero's results on selected chart and OCR tasks in the main text, Tab. 9 illustrates a comprehensive evaluation across an extended set of tasks. Notably, VisionZero-Qwen-7B consistently surpasses baseline methods across diverse OCR and chart-based tasks. Particularly, VisionZero-Qwen-7B (Chart) exhibits superior performance and significant capability enhancement due to its targeted training on chart datasets. For example, on the InfoVQA benchmark, VisionZero-Qwen-7B (Chart) improved the performance of the original model by approximately 4%, outperforming the state-of-the-art ViGal by 14%. This substantial improvement arises because baselines trained extensively on reasoning datasets typically suffer from task overfitting, whereas Vision-Zero circumvents this limitation by concurrently fostering multiple capabilities.

Table 6: Performance comparison between Vision-Zero and other models on OCR, Chart, and Document Understanding. All models are evaluated using the open-source platform VLMEvalKit.

| Model | AI2D | ChartQA | TextVQA | DocVQA | InfoVQA | OCR Bench | SEEDBench2 |
|---|---|---|---|---|---|---|---|
| *Proprietary Model* | | | | | | | |
| GPT4o | 84.4 | 85.7 | 82.2 | 91.1 | 78.2 | 73.9 | 72.0 |
| *Performance on Qwen2.5-VL-7B* | | | | | | | |
| Qwen2.5-VL-7B-Instruct | 84.7 | 86.1 | 85.5 | 94.8 | 82.3 | 88.3 | 70.4 |
| R1-OneVision-7B | 82.2 | – | – | – | – | 81.0 | 66.4 |
| MM-Eureka-Qwen-7B | 84.1 | 77.3 | 81.1 | 81.1 | 71.7 | 86.7 | 68.2 |
| VLAA-Thinker-7B | 84.0 | 84.3 | 82.9 | 92.7 | 71.8 | 86.9 | 67.4 |
| OpenVLThinker-7B | 81.8 | – | – | – | – | 83.3 | 68.0 |
| ViGaL-Snake+Rotation | 84.5 | 79.9 | 82.2 | 92.5 | 72.7 | 86.8 | 69.1 |
| **VisionZero-Qwen-7B (CLEVR)** | **84.5** | **86.3** | **85.3** | **94.9** | **82.5** | **88.1** | **69.5** |
| **VisionZero-Qwen-7B (Chart)** | **85.8** | **87.2** | **86.4** | **95.9** | **86.5** | **89.0** | **70.9** |
| **VisionZero-Qwen-7B (Real-World)** | **84.8** | **86.3** | **85.4** | **95.2** | **82.3** | **88.5** | **69.8** |

### A.4.2 COMPREHENSIVE EVALUATION ON VISION-CENTRIC TASKS

Moreover, as shown is Tab. 7, Vision-Zero achieves top-tier performance across six distinct vision-centric task groups. VisionZero-Qwen-7B (CLEVR), whose training data has stronger visual emphasis compared to VisionZero-Qwen-7B (Chart), obtains even better results. Specifically, VisionZero-Qwen-7B (CLEVR) surpasses state-of-the-art baselines by 1.1% on average across the six task categories. These results underscore the potential and applicability of Vision-Zero as the first zero-human-in-the-loop training paradigm.

Table 7: Performance comparison between Vision-Zero and other state-of-the-art models on Vision-Centric benchmarks. All models are evaluated using the open-source platform VLMEvalKit.

| Model | RealworldQA | MMVP | MMStar | BLINK | MuirBench | CRPE | Avg. |
|---|---|---|---|---|---|---|---|
| *Proprietary Model* | | | | | | | |
| GPT4o | 75.4 | 86.3 | – | 68.0 | 68.0 | – | – |
| *Performance on Qwen2.5-VL-7B* | | | | | | | |
| Qwen2.5-VL-7B-Instruct | 68.1 | 76.8 | 64.6 | 55.2 | 58.2 | 76.4 | 66.6 |
| R1-OneVision-7B | 58.0 | 61.3 | 57.8 | 48.7 | 46.3 | 75.3 | 57.9 |
| MM-Eureka-Qwen-7B | 66.1 | 74.3 | **65.9** | 54.0 | **61.1** | **76.7** | 66.4 |
| VLAA-Thinker-7B | 65.4 | 71.6 | 60.4 | 53.0 | 57.1 | 74.6 | 63.7 |
| OpenVLThinker-7B | 60.2 | 71.3 | 59.1 | 49.9 | 52.8 | 75.8 | 61.5 |
| ViGaL-Snake+Rotation | 66.5 | 74.6 | 62.6 | 55.6 | 57.9 | 76.7 | 65.7 |
| **VisionZero-Qwen-7B (CLEVR)** | **68.5** | **79.2** | **65.2** | **57.2** | **59.4** | **76.9** | **67.7** |
| **VisionZero-Qwen-7B (Chart)** | **68.2** | **77.9** | 64.7 | **56.1** | 58.6 | 76.2 | **66.9** |
| **VisionZero-Qwen-7B (Real-World)** | **68.5** | **79.5** | **65.8** | **57.5** | **59.8** | **77.0** | **68.0** |

### A.4.3 COMPARISON WITH CONTRASTIVE RLVR

To evaluate the advantages of Vision-Zero over prior contrastive RLVR methods in enhancing VLM reasoning capabilities, we conducted a direct comparison under identical training conditions with MiCo (Chen et al., 2025b).

Specifically, to align with the MiCo-7B setup, we trained Vision-Zero on the OmniEdit dataset using the Qwen2.5-VL-7B model for 100 iterations with a batch size of 128. In contrast, MiCo-7B was trained for 600 iterations with the same batch size. Although Vision-Zero adopts a multi-round training paradigm, its overall training cost remains comparable. As shown in Tab. 8, we evaluated Vision-Zero's reasoning performance on six benchmark datasets consistent with those used for MiCo-7B, with MiCo's results taken directly from its original paper.

Table 8: Performance Comparison of Qwen2.5VL-7B and MiCo-7B across multiple benchmarks.

| Model | MuirBench | Blink | Hallusion | MMStar | MMMU | MathVista |
|---|---|---|---|---|---|---|
| Qwen2.5VL-7B | 58.4 | 55.5 | 69.5 | 64.1 | 54.1 | 67.1 |
| MiCo-7B | 60.5 | 57.2 | 69.6 | 65.6 | 54.8 | 67.9 |
| VisionZero-Qwen-7B(OminiEdit) | **62.4** | **58.9** | **71.2** | **66.2** | **55.7** | **69.1** |

The results demonstrate that, whereas MiCo is primarily optimized for multi-image difference reasoning, Vision-Zero benefits from a strategically constructed training environment that better targets the development of reasoning skills. As a result, Vision-Zero consistently outperforms MiCo across all six general reasoning benchmarks. This indicates that the combination of a self-play game mechanism and a strategic interaction environment enables Vision-Zero to equip the model with significantly stronger general-purpose reasoning capabilities than contrastive RLVR approaches.

### A.4.4 COMPARISON WITH CHART-SPECIALIZED MODELS

To thoroughly evaluate whether Vision-Zero can enhance a model's chart understanding capabilities and establish its superiority in the chart reasoning domain, we conduct a dedicated comparison against models specifically fine-tuned for chart understanding, including ECD (Yang et al., 2025a), Bespoke-MiniChart-7B (Tang et al., 2025) and Chart-R1-7B (–, 2025). We compare the number of chart images, the number of QA annotations used for training, and the final performance on standard chart understanding benchmarks.

Table 9: Performance Comparison of VisionZero-Qwen-7B(chart) and chart-specialized models across multiple benchmarks.

| Model | Chart Number | QA Number | ChartXiv_RQ | ReachQA | Avg. |
|---|---|---|---|---|---|
| Qwen2.5VL-7b | - | - | 42.5 | 50.8 | 46.7 |
| ECD-Qwen2.5VL-7b | 10.5k | 320k | 40.2 | 53.5 | 46.9 |
| Bespoke-MiniChart | 13.0k | 91k | 46.2 | **54.0** | 50.1 |
| Chart-R1-7B | 21.9k | 32.7k | 46.2 | - | - |
| VisionZero-Qwen-7B(chart) | 2.0k | 0 | **46.6** | 53.8 | **50.2** |

As shown in the Tab. 9, VisionZero-Qwen-7B (Chart) achieves comparable or superior performance to models trained on large-scale chart-specific data—without requiring any QA supervision. Notably, on ChartXIV_QA, VisionZero-Qwen-7B delivers a 4.1% improvement, surpassing models that were explicitly trained on chart datasets. These results highlight Vision-Zero's ability to generalize across domains and demonstrate its potential as a cost-effective and competitive training framework for enhancing model performance in diverse visual reasoning tasks.

### A.4.5 COMPARISON OF CoTs ON GENERAL QA TASK BEFORE AND AFTER TRAINING.

To intuitively demonstrate how the "Who is Spy" game enhances model performance on general QA tasks, we present illustrative examples comparing model thinking before and after training on identical general QA tasks. As shown in the boxes below, the trained model clearly demonstrates improved thought processes and logical reasoning, explicitly understanding the steps required. Additionally, it more accurately focuses on critical information in images, exhibiting enhanced key content-capturing abilities, directly leading to more accurate responses.

This improvement arises from two primary factors: first, the capabilities required by the "Who is Spy" game closely overlap with those needed for general QA tasks. For instance, during the clue-giving stage, the model must identify differences between images and text, directly training fine-grained visual comparison and comprehension skills. Second, the strategic and challenging environment of the "Who is Spy" game encourages models to cultivate meticulous reasoning and logical thinking habits during responses, further contributing to improved performance in general QA tasks.

---

**Example1: Comparison of CoTs Before and After Training**

**Question:** How many countries have people working for more than 35 hours over the years?

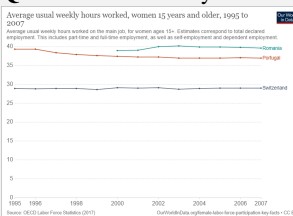

**Untrained Model:** The graph compares the average usual weekly hours worked by women over the age of 15 across three countries (Romania, Portugal, and Switzerland) from 1995 to 2007. Each country is represented by a different colored line: Romania is blue, Portugal is orange, and Switzerland is dark blue. From 1995 to 2007, the line representing Romania (blue) is consistently above the 35-hour mark on the y-axis, which measures average usual weekly hours. The lines for Portugal and Switzerland are below the 35-hour mark throughout this period. Thus, only one country, Romania, shows average weekly hours worked exceeding 35 hours over the entire period displayed.

**Trained Model:** To determine how many countries have people working for more than 35 hours on average weekly over the years from 1995 to 2007, let's analyze the data shown in the graph step by step:
1. **Identify the countries represented:** - Romania (in blue) - Portugal (in orange) - Switzerland (in light blue)
2. **Analyze the hours worked for each country:** - **Romania:** The blue line representing Romania shows that the average usual weekly hours worked remain consistently above 35 hours throughout the entire period from 1995 to 2007. - **Portugal:** The orange line representing Portugal shows that the average usual weekly hours worked also remain consistently above 35 hours over the same period. - **Switzerland:** The light blue line representing Switzerland shows that the average usual weekly hours worked remain consistently below 35 hours over the entire period.
3. **Determine the countries with averages above 35 hours:** - From the above analysis, Romania and Portugal have average usual weekly hours worked consistently above 35 hours. - Switzerland has average usual weekly hours worked consistently below 35 hours.
Therefore, the number of countries with people working for more than 35 hours on average weekly over the years is **2**.

---

**Example2: Comparison of CoTs Before and After Training**

**Question:** How many bricks are missing in the wall?

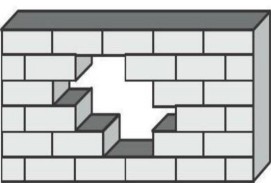

**Untrained Model:** To determine how many bricks are missing in the wall, we need to count the number of bricks that are not present in the wall structure. Let's analyze the image step by step.
1. **Identify the pattern of the missing bricks:** - The missing bricks form a zigzag pattern from the top left to the bottom right of the wall.
2. **Count the missing bricks in each row:** - **Top row:** 1 brick is missing. - **Second row:** 2 bricks are missing. - **Third row:** 3 bricks are missing. - **Fourth row:** 2 bricks are missing. - **Bottom row:** 1 brick is missing.
3. **Sum the number of missing bricks:** - Add the number of missing bricks from each row: 1 + 2 + 3 + 2 + 1 = 9 Therefore, the total number of bricks missing in the wall is 9.

**Trained Model:** To find out how many bricks are missing in the wall, we need to count the voids where bricks should be but aren't.
1. Count the missing bricks in the top row: 1 brick. 2. Count the missing bricks in the second row: 2 bricks. 3. Count the missing bricks in the third row: 2 bricks. 4. Count the missing bricks in the fourth row: 1 brick.
Adding these up, we get a total of 1 + 2 + 2 + 1 = 6 bricks missing.

### A.4.6 STABILITY ANALYSIS OF EDITOR CAPABILITIES

To assess whether Vision-Zero heavily depends on high-performance image editors, we introduced 20% noise into the RealWorld dataset to simulate scenarios of partial editing failures. Specifically, we replaced 10% of the modified images (provided to the spy) with blank images and another 10% with images identical to the originals (provided to civilians). We then trained the model on this noisy dataset and evaluated its performance on the validation set.

The experimental results, summarized in the Tab. 10, indicate that Vision-Zero maintains performance improvements even with partially failed edits. Although the improvement is slightly lower compared to training without noise, the model still achieves an average accuracy increase of 2.8% over the baseline. This demonstrates Vision-Zero's strong robustness to editing failures, highlighting that it can enhance model performance without relying heavily on sophisticated image editors.

Table 10: Performance of the model trained on dataset containing images that failed to be edited.

| Model | Mathvista | MathVison | WeMath | MathVerse | LogicVista | DynaMath | Avg. |
|---|---|---|---|---|---|---|---|
| Qwen2.5VL-7b | 68.2 | 25.4 | 36.1 | 49.0 | 47.2 | 20.9 | 41.1 |
| VisionZero-Qwen-7B (Real-World) | 72.4 | 28.0 | 39.5 | 52.2 | 50.3 | 22.1 | 44.1 |
| **VisionZero-Qwen-7B (Real-World+Noise)** | 72.2 | 27.5 | 39.2 | 52.4 | 50.1 | 21.9 | 43.9 |

We also present the model's responses to failed image edits (when the spy input is a blank image) in the block below. It can be observed that even when faced with failed edits, the model continues to reason effectively, leveraging alternative available information to maximize rewards. Thus, even image pairs resulting from unsuccessful edits remain valuable training samples for the model.

---

**Example3: Reasoning of the model when editing failure (the spy obtained a blank image)**

**Civ1:** The red cube is in front of the purple cylinder.
**Civ2:** There are two small cubes, one blue and one purple, near the purple cylinder and the red cube.
**Civ3:** The purple cylinder is behind the red cube and the two small cubes.

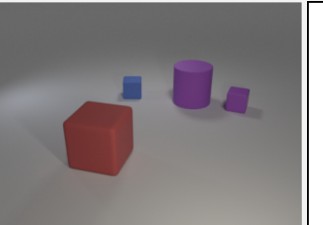

**Spy:** - **Player 1's clue**: The red cube is in front of the purple cylinder. - **Player 2's clue**: There are two small cubes, one blue and one purple, near the purple cylinder and the red cube. - **Player 3's clue**: The purple cylinder is behind the red cube and the two small cubes.
Given that I am the SPY, my image has two different objects from the others, but their positions remain the same. Based on the clues:
- The common elements seem to be: - A red cube - A purple cylinder - A blue small cube - A purple small cube
- My differences could be: - The color or shape of one of the small cubes (blue or purple) - Possibly the position or size, but as per rules, only color and specific features should be described.
Since descriptions so far mention color and proximity, I should maintain consistency and describe features that align with their clues to avoid suspicion.
Description: The red cube is in front of the purple cylinder and the two small cubes.

---

### A.4.7 PARAMETER AND MODULE ABLATION

In this section, we conduct ablation studies on the key parameters introduced in Vision-Zero, including the number of civilian players, the number of clue rounds, and the Role-Advantage Estimation (RAE) module, to analyze the robustness of Vision-Zero and the contribution of each component.

We first evaluate whether Vision-Zero can improve model performance under different numbers of players. We train models with 2, 3, and 4 civilian players (with the spy always fixed to one) and

report the results in the Tab. 11. Vision-Zero consistently improves model performance across all player settings, demonstrating its robustness to variations in player count. As the number of players increases, the game environment becomes more challenging, resulting in more reasoning steps and reward signals per sample, thus yielding greater performance gains. Even in the minimal two-player setting, the model still achieves an average improvement of 1.3% across six benchmarks.

Table 11: Performance of the model under different numbers of players.

| Model | Mathvista | MathVison | WeMath | MathVerse | LogicVista | DynaMath | Avg. |
|---|---|---|---|---|---|---|---|
| Qwen2.5VL-7b | 68.2 | 25.4 | 36.1 | 49.0 | 47.2 | 20.9 | 41.1 |
| **VisionZero-Qwen-7B (CLEVER Civ=2)** | 69.7 | 26.5 | 37.2 | 50.6 | 48.9 | 21.2 | 42.4 |
| **VisionZero-Qwen-7B (CLEVER Civ=3)** | 72.6 | 28.1 | 39.8 | 51.9 | 50.1 | 22.3 | 44.1 |
| **VisionZero-Qwen-7B (CLEVER Civ=4)** | 73.2 | 28.0 | 40.0 | 52.1 | 51.8 | 24.1 | 44.9 |

We also examine the effect of varying the number of clue rounds in each game by training models with 1, 2, and 3 clue rounds. As shown in the Tab. 12, the performance gain is small when only a single clue round is used, due to insufficient information available for the decision stage and limited opportunities for multi-step integration during training. In contrast, using two or three clue rounds leads to notable improvements, achieving average gains of 3% and 4.1% across six benchmarks. As the number of clue rounds increases, the model is required to process and integrate more information, resulting in progressively stronger performance.

Table 12: Performance of the model under different round numbers of clue stage.

| Model | Mathvista | MathVison | WeMath | MathVerse | LogicVista | DynaMath | Avg. |
|---|---|---|---|---|---|---|---|
| Qwen2.5VL-7b | 68.2 | 25.4 | 36.1 | 49.0 | 47.2 | 20.9 | 41.1 |
| **VisionZero-Qwen-7B (CLEVER Clue_Rd = 1)** | 68.0 | 26.5 | 36.5 | 48.2 | 47.2 | 21.2 | 41.3 |
| **VisionZero-Qwen-7B (CLEVER Clue_Rd = 2)** | 72.6 | 28.1 | 39.8 | 51.9 | 50.1 | 22.3 | 44.1 |
| **VisionZero-Qwen-7B (CLEVER Clue_Rd = 3)** | 73.1 | 29.0 | 40.5 | 52.2 | 52.0 | 24.3 | 45.2 |

To assess the importance of the RAE module, we compare models trained with and without RAE. In the no-RAE setting, we update the model weights directly based on the reward after the clue stage without subtracting the role-advantage baseline. The results in the Tab. 13 show that removing RAE leads to negative performance gains. This occurs because spies and civilians inherently possess asymmetric information due to differences in their assigned images and roles. As a result, the game can be intrinsically easier or harder depending on the role. Without adjusting for role advantages, directly backpropagating rewards based on win/loss signals fails to reflect the model's true performance level, preventing effective learning. These findings highlight the critical importance of the RAE module in Vision-Zero.

Table 13: Performance of the model under w/ and w/o RAE module.

| Model | Mathvista | MathVison | WeMath | MathVerse | LogicVista | DynaMath | Avg. |
|---|---|---|---|---|---|---|---|
| Qwen2.5VL-7b | 68.2 | 25.4 | 36.1 | 49.0 | 47.2 | 20.9 | 41.1 |
| **VisionZero-Qwen-7B (CLEVER w/ RAE)** | 72.6 | 28.1 | 39.8 | 51.9 | 50.1 | 22.3 | 44.1 |
| **VisionZero-Qwen-7B (CLEVER w/o RAE)** | 65.2 | 21.3 | 30.1 | 47.2 | 44.3 | 16.1 | 37.4 |

## A.5 LIMITATION AND FUTURE WORK

In this section, we discuss potential limitations of Vision-Zero and outline directions for future research. Firstly, the implementation of Vision-Zero relies on image editors to produce differentiated image pairs. Consequently, its application might be limited in highly specialized or resource-constrained domains, such as medical imaging, scientific charts, and remote sensing, where such edited data might not be readily available. Secondly, the current framework is designed around single-image observations and pairwise edits. Extending this framework to richer modalities, including extended videos, complex multi-image contexts, or interactive 3D environments, may require significant redesign of the game mechanics and training algorithms. Addressing these limitations constitutes an essential direction for future work.

## A.6 THE USE OF LARGE LANGUAGE MODELS

In this work, we used ChatGPT-4o (OpenAI) and Gemini 2.5 Flash (Google) to assist with image generation for dataset construction. Specifically, the models were prompted to edite visual content used in training datasets. We gratefully acknowledge their utility in facilitating efficient data synthesis.

