# OpenReview forum: "Vision-Zero: Scalable VLM Self-Evolution via Multi-Agent Self-Play"
_ICLR.cc/2026/Conference — ICLR 2026 Poster_

### Official Review · Reviewer_B6Um · 2025-10-18

**Soundness:** 2
**Presentation:** 2
**Contribution:** 2
**Rating:** 4
**Confidence:** 5

**Summary:**

The paper proposes Vision-Zero, a zero-human-in-the-loop post-training framework for VLMs built upon a visual “Who is the Spy?” self-play game. This paper further proposes iterative self-play optimization for this self-play game task, which consists of self-play optimization in clue stage and RLVR for the decision stage. Experiment demonstrate consistent gains over baseline methods for Qwen2.5-VL-7B base model.

**Strengths:**

1.	This paper demonstrates that self-play visual game can be an effective task for VLM post-training, which is an interesting finding.

2.	The paper evaluates on a reasonably wide set of benchmarks to demonstrate the effectiveness of Vision-Zero.

**Weaknesses:**

1.	Prior work has already explored automatically crafted multi-image contrastive data for VLM post-training (e.g., MiCo). The paper should avoid claiming to be the first zero-human-in-the-loop paradigm and more carefully situate its contribution relative to [1].

2.	The “Who is the Spy?” pipeline requires multiple forward passes per sample and a more complex training algorithm. To justify this complexity, the paper should include direct, apples-to-apples comparisons against simpler multi-image contrast baselines such as MiCo [1], using the same data sources (and ideally matched compute) to test whether the game mechanics yield additional gains beyond contrastive RLVR.

3.	Please report training efficiency for Vision-Zero. The multi-turn interactions per sample could introduce substantial overhead; quantifying this is important for practical adoption.

4.	The main text states 100 iterations while the appendix reports 150. Please reconcile this discrepancy and clarify which results correspond to which training steps.

[1] MiCo: Multi-image Contrast for Reinforcement Visual Reasoning. NeurIPS 2025

**Questions:**

Please see Weaknesses

---

> ### Author Response · Authors · 2025-11-25
> **Response to Reviewer B6Um [Part1]**
>
> We would like to thank reviewer B6Um for taking the time to review our paper and provide valuable feedback. We are thankful to have the chance to address the questions and concerns. These edits will make the paper stronger.
>
> ## **W1. Avoid claiming to be the first zero-human-in-the-loop paradigm and more carefully situate its contribution relative to MiCo.**
>
> **Responce:** We thank the reviewer for bringing up MiCo and fully agree that our contribution should be more carefully positioned with respect to it.
>
> MiCo also follows a zero-human-in-the-loop philosophy, it automatically constructs “similar but different” image triplets from videos and image-editing datasets, and trains VLMs with same/different questions plus rule-based RL to perform fine-grained multi-image comparison, without any human-annotated QA pairs. It is a promising approach for strengthening multi-image, fine-grained visual comparison.
>
> Compared with MiCo, our approach follows a substantially different motivation and methodological design, with the key distinctions and contributions reflected in the following two aspects:
>
> - **Difference in Environment Setup.** MiCo places the model in a setting where it performs reasoning over a fixed set of fully observable images. Once a set of samples are determined, the environment is static and does not change in one episode. In contrast, Vision-Zero operates within a zero-sum self-play adversarial environment, where each agent (i.e., civilian or spy) receives incomplete information. Every action taken by an agent becomes part of the environment for the other agents. As agents become stronger, they naturally create more challenging environments for one another. This dynamic loop forces each agent to continually improve its reasoning ability to handle the increasing complexity, enabling sustainable self-improvement of the model. In summary, compared with MiCo, Vision-Zero features: ***(a) Incomplete-information reasoning***.   ***(b) More complex strategic reasoning***.  ***(c) Agents dynamically generating environments for one another.***  ***(d) Adaptively increasing environment complexity.***
> - **Difference in Target Tasks.** MiCo focuses on strengthening the model’s ability to reason about visual differences and consistencies across multiple images, enabling it to detect subtle variations. In contrast, Vision-Zero is designed to improve general-purpose visual understanding across a broad spectrum of tasks. Its gamified self-play framework naturally accommodates diverse image types including natural images, synthesized scenes, document images and charts etc. Within the game, the model must interpret these visuals, extract salient features, articulate meaning clues, and engage in strategic inference, thereby learning capabilities that transfer to a broad range of visual tasks.
>
> In response to the reviewer’s suggestion, **we have revised the manuscript** to avoid claiming to be the first zero-human-in-the-loop paradigm in general. Instead, we now state that Vision-Zero is “the first gamified self-play framework for VLMs that achieves zero-human-in-the-loop post-training.” .We have also added a paragraph in **Appendix~A.1** (background) that discusses MiCo and highlights these differences.

---

> ### Author Response · Authors · 2025-11-25
> **Response to Reviewer B6Um [Part2]**
>
> ## **W2. Need comparisons against simpler multi-image contrast baselines such as MiCo.**
>
> **Responce:** We appreciate the reviewer’s suggestion to include direct comparisons against simpler contrastive RLVR methods. In response, we trained Vision-Zero under the same settings as MiCo and directly compared the experimental results.
>
> **[Experiment Setting]** To ensure a fair comparison with MiCo-7B, we follow its training configuration by using Qwen2.5-VL-8B as the base model and training on the OmniEdit dataset with a global batch size of 128. In the original MiCo paper, MiCo-7B is trained for 600 iterations under this setup. For Vision-Zero-7B, we adopt the same batch size and train for 100 iterations. Despite our method involving multiple rounds, the overall data and training budget remains comparable. A more detailed efficiency analysis of Vision-Zero is provided in our response to W3.
>
> **[Result]** The experimental results, presented in the table below, evaluate the reasoning capabilities of the Vision-Zero model across six benchmarks consistent with those used by MiCo-7B, whose results are cited directly from their original paper.
>
> | Model                             | MuirBench | Blink    | Hallusion | MMStar   | MMMU     | MathVista |
> | --------------------------------- | --------- | -------- | --------- | -------- | -------- | --------- |
> | Qwen2.5VL-7B                      | 58.4      | 55.5     | 69.5      | 64.1     | 54.1     | 67.1      |
> | MiCo-7B                           | 60.5      | 57.2     | 69.6      | 65.6     | 54.8     | 67.9      |
> | **VisionZero-Qwen-7B (OmniEdit)** | **62.4**  | **58.9** | **71.2**  | **66.2** | **55.7** | **69.1**  |
>
> As shown, while MiCo primarily focuses on multi-image difference detection, Vision-Zero benefits from a strategically designed training environment that more directly targets the development of reasoning capabilities in the model. Consequently, Vision-Zero achieves consistently superior performance across a range of general reasoning benchmarks, outperforming MICO on all six benchmarks. These results suggest that, owing to its self-play game mechanism and strategic training setup, Vision-Zero enables the model to acquire stronger general-purpose reasoning abilities compared to contrastive RLVR approaches.
>
> We have added these experimental results and a detailed comparison between Vision-zero and contrastive RLVR in **Appendix~ 4.3**.

---

> ### Author Response · Authors · 2025-11-25
> **Response to Reviewer B6Um [Part3]**
>
> ## **W3. Quantify training efficiency for Vision-Zero.**
>
> **Responce:** We agree with the reviewer that it is important to more precisely quantify the training efficiency of Vision-Zero, and we appreciate the concern regarding the potential training overhead introduced by its multi-round interaction mechanism.
>
> **[Why Vision-Zero does not incur prohibitive overhead.]**
>
> We first clarify why Vision-Zero does **not** incur significant additional computational cost:
>
> First, Vision-Zero adopts a **fixed interaction pattern** (two clue rounds + one decision round), such that all games in the same batch share an identical three-stage computation graph. This design enables **fully parallelized** forward and backward passes across multiple games within a batch, without any asynchronous waiting or gradient interference between games. In practice, the GPU utilization is comparable to that of GRPO.
>
> Second, each Vision-Zero game generates **multiple actionable steps**, resulting in **denser learning signals** and improved sample efficiency. For a game involving n_c civilians and one spy, a single episode generates (n_c+1)*(2 clue rounds + 1 decision round) actions, each associated with verifiable rewards and losses. This provides more supervision per sample than standard GRPO.
>
>
>
> **[Experiment 1: Wall-clock training time vs. vanilla GRPO.]**
>
> To provide a concrete quantification, we first compare Vision-Zero with the vanilla GRPO under same hardware and batch settings. For vanilla GRPO, we use MM-Eureka training dataset and train Qwen2.5-VL-7B and InternVL3-8B for 100 iterations on 8 NVIDIA A100 (80GB) GPUs with a global batch size of 128. For Vision-Zero, we use the same machines and batch size, train for 100 iterations on the CLEVR-based dataset, and measure both wall-clock training time and downstream performance.
>
> We record the total training time and evaluate checkpoints at different iterations on MathVision. The results (as Table below, Up is Qwen2.5VL-7B, Down is InternVL3-8B) show that, although a single iteration of Vision-Zero takes about ∼2× longer than a single GRPO iteration (due to the multi-round interactions), Vision-Zero achieves much higher overall training efficiency: it is **2.67×** more time-efficient on Qwen 2.5-VL-7B and **6.40×** more time-efficient on InternVL3-8B in terms of performance gain per iteration, and also yields higher final accuracy on the validation set. This indicates that the multi-turn interactions do not lead to prohibitive training cost; instead, they improve sample efficiency and convergence speed.
>
> | Model                         | 0    | 10   | 20   | 30   | 40   | 50   | 60   | 70   | 80   | 90   | 100  |
> | ----------------------------- | ---- | ---- | ---- | ---- | ---- | ---- | ---- | ---- | ---- | ---- | ---- |
> | MM-Eureka-Qwen-7B(GRPO)             | 25.4 | 25.2 | 25.6 | 25.4 | 25.8 | 26.3 | 26.5 | 26.5 | 26.7 | 26.5 | 26.7 |
> | **VisionZero-Qwen-7B(CLEVR)** | 25.4 | 25.6 | 26.1 | 26.8 | 27.2 | 27.1 | 27.6 | 27.9 | 27.7 | 27.9 | 28.1 |
>
>
> | Model                         | 0    | 10   | 20   | 30   | 40   | 50   | 60   | 70   | 80   | 90   | 100  |
> | ----------------------------- | ---- | ---- | ---- | ---- | ---- | ---- | ---- | ---- | ---- | ---- | ---- |
> | MM-Eureka-InternVL3-8B(GRPO)              | 21.3 | 21.0 | 21.5 | 21.8 | 21.6 | 22.2 | 22.4 | 22.2 | 22.5 | 22.2 | 22.5 |
> | **VisionZero-InternVL3-8B(CLEVR)** | 21.3 | 22.4 | 23.0 | 23.2 | 23.6 | 23.4 | 24.0 | 23.8 | 24.2 | 24.0 | 24.2 |
>
>
> | Model                        | Time             | Performance |
> | ---------------------------- | ---------------- | ----------- |
> | MM-Eureka-Qwen-7B(GRPO)         | 60.5 A100-Hours  | 26.7        |
> | **VisionZero-Qwen2.5-VL-7B(CLEVR)** | 126.5 A100-Hours | **28.1**    |
> | MM-Eureka-InternVL3-8B(GRPO)             | 68.5 A100-Hours  | 22.5        |
> | **VisionZero-InternVL3-8B(CLEVR)**  | 138 A100-Hours   | **24.2**    |

---

> ### Author Response · Authors · 2025-11-25
> **Response to Reviewer B6Um [Part4]**
>
> **[Experiment 2: Training cost vs. SOTA RL baselines.]**
>
> We further compare Vision-Zero against SOTA RL-based VLM post-training methods in terms of training cost. For each baseline, we extract from the original paper the number of RL training samples used, and multiply it by the standard GRPO per-sample cost to obtain an approximate RL training time under a fully fair setting (this estimate is conservative as it ignores the additional SFT cost for baselines). The result is shown in Table below.As shown, Vision-Zero remains highly competitive in both sample efficiency and total training time: it achieves higher performance while using a similar or even smaller estimated RL budget. It attributes to the challenging interactive training environment and the dense reward structure of Vision-Zero.
>
> | Method                         | Num (RL) | Label Cost (Tokens) | Training Method              | Interact | Time Cost          | MMMU     | MMMU_pro |
> | ------------------------------ | -------- | ------------------- | ---------------------------- | -------- | ------------------ | -------- | -------- |
> | Qwen2.5-VL-7B                  | –        | –                   | –                            | –        | –                  | 54.3     | 37.0     |
> | R1-OneVision-7B                | 10k      | > 1.1M              | SFT+GRPO                     | ✗        | ≥ 170 A100-Hours   | 51.9     | 32.6     |
> | VLAA-Thinker-7B                | 25k      | 29.6M               | SFT+GRPO                     | ✗        | ≥ 120 A100-Hours   | 48.2     | 31.9     |
> | OpenVLThinker-7B               | 10k      | 5.7M                | SFT+GRPO                     | ✗        | ≥ 125 A100-Hours   | 54.8     | 22.1     |
> | MM-Eureka-Qwen-7B              | 15k      | –                   | GRPO                         | ✗        | ≈ 700 A100-Hours   | 55.8     | 36.9     |
> | ViGaL-Snake                    | –        | 0                   | RLOO                         | ✗        | ≈ 170 A100-Hours   | 55.8     | 36.6     |
> | **VisionZero-Qwen-7B (CLEVR)** | **2k**   | **0**               | Alternating Self-play + GRPO | **✔**    | **127 A100-Hours** | **58.8** | **37.7** |
>
>
>
> We have added the above experiments and corresponding explanations to **Section 3.1** of the paper to further clarify Vision-Zero's training efficiency and cost.
>
>
>
> ## **W4. Inconsistencies in iteration.**
>
> **Response:** We thank the reviewer for catching this inconsistency. The correct number of training iterations for all experiments is 100. The mention of 150 iterations in the Appendix was a typo.
> We have corrected this in the revised version and ensured that all references to the training setup are consistent.

---

### Official Review · Reviewer_K9Zp · 2025-10-27

**Soundness:** 3
**Presentation:** 3
**Contribution:** 2
**Rating:** 6
**Confidence:** 4

**Summary:**

This paper introduces Vision-Zero, a framework inspired by the “Spy vs. Player” game concept. The idea is creative and brings an interesting self-play strategy from LLMs into the vision-language domain (VLMs). The design is simple yet somewhat innovative. However, the overall performance improvement is modest and appears constrained by the capability of the underlying image editing models.

**Strengths:**

1.	The paper identifies two valuable and timely challenges in multimodal learning: data scarcity and knowledge ceiling.
2.	It presents a clear motivation by extending the concept of self-play from large language models (LLMs) to vision-language models (VLMs).
3.	The framework generates data using automated image editing tools or procedural rendering, which supports scalable dataset creation.
4.	The Spy–Player game formulation is an interesting and original idea that provides a novel self-supervised interaction mechanism.

**Weaknesses:**

1.	The paper identifies two valuable and timely challenges in multimodal learning: data scarcity and knowledge ceiling.
2.	It presents a clear motivation by extending the concept of self-play from large language models (LLMs) to vision-language models (VLMs).
3.	The framework generates data using automated image editing tools or procedural rendering, which supports scalable dataset creation.
4.	The Spy–Player game formulation is an interesting and original idea that provides a novel self-supervised interaction mechanism.

**Questions:**

1.	How fair is the comparison with prior works that do not assume access to camera poses? How does MultiViewPano perform when camera poses are inaccurate or unavailable?
2.	How robust is the proposed method to poor-quality SEVA outputs? Can the authors provide visualizations of such cases?
3.	Could the authors expand the ablation study and visualize intermediate outputs to clarify which modules contribute to which improvements?
4.	How are geometric distortions and artifacts handled during repeated enhancement steps?
5.	Why does the proposed method not outperform CubeDiff, especially for indoor scenes?
6.	Could the authors ensure that all best-performing metrics in Table 1 are boldfaced for readability?

---

> ### Author Response · Authors · 2025-11-25
> **Response to Reviewer K9Zp [Part1]**
>
> We would like to thank the reviewer for taking the time to review our paper and provide valuable feedback. We are glad that the reviewer thinks our paper is clear motivation and novel. We are excited to have the chance to address the reviewer’s questions and concerns. These edits will make the paper stronger.
>
> ## **W1. Questionable benefit over training on images generated via image editing or procedural rendering.**
>
> **Response:** We thank the reviewer for highlighting the comparison between Vision-Zero and approaches directly trained on images generated via image editing or procedural rendering, as both methodologies support zero-human-in-the-loop scenarios. In response, we address the differences and advantages of Vision-Zero through methodological explanations and empirical comparisons.
>
>
>
> **[Advantages of Vision-Zero Over Direct Training on Edited or Procedurally Rendered Images.]**
>
>  We acknowledge the reviewer’s concern and have reviewed relevant literature. Compared to these methods, Vision-Zero provides two distinct advantages:
>
> - **Vision-Zero introduces a self-play environment** (i.e., the spy and civilians have zero-sum rewards). As the model's capabilities improve, the game environment naturally becomes more challenging, enabling continuous performance enhancement. This paradigm effectively prevents the knowledge saturation problem encountered by traditional RL methods when trained solely on static data, offering a framework that allows unlimited model learning potential.
> - **Vision-Zero specifically aims to improve the general reasoning abilities of VLMs,** rather than narrowly focusing on detailed image comparison or perceptual capabilities. During training, Vision-Zero uses the model's own high-quality reasoning outputs as training data. To succeed, the model must comprehensively integrate statements from other players and strategically mislead opponents. Consequently, achieving high rewards demands strong capabilities in planning, retrieval, decomposition, and strategizing (as illustrated in Figure 5 of the main text), explicitly targeting general reasoning skills.
>
>
>
> **[Experiment: Comparative Performance Against Direct Training on Edited Images.]**
>
> To further demonstrate Vision-Zero's advantages, we directly compared it to MiCo (which also adheres to a zero-human-in-the-loop philosophy by automatically constructing "similar-but-different" image triplets from video frames and image-editing datasets. MiCo utilizes fine-grained comparative training across multiple images using same/different contrastive questions and rule-based RL). Using identical training settings (same datasets and models), we evaluated models trained with Vision-Zero and MiCo on several reasoning benchmarks. The experimental results presented below clearly show that Vision-Zero consistently achieves superior performance improvements across multiple general reasoning benchmarks.
>
> | Model                             | MuirBench | Blink    | Hallusion | MMStar   | MMMU     | MathVista |
> | --------------------------------- | --------- | -------- | --------- | -------- | -------- | --------- |
> | Qwen2.5VL-7B                      | 58.4      | 55.5     | 69.5      | 64.1     | 54.1     | 67.1      |
> | MiCo-7B                           | 60.5      | 57.2     | 69.6      | 65.6     | 54.8     | 67.9      |
> | **VisionZero-Qwen-7B (OmniEdit)** | **62.4**  | **58.9** | **71.2**  | **66.2** | **55.7** | **69.1**  |
>
>
>
> This strongly indicates that, due to its self-play mechanism and strategically complex game environment, Vision-Zero substantially enhances the general reasoning capabilities of VLMs compared to direct training on edited images.
>
> We have included detailed results of this experiment and further discussions comparing Vision-Zero with direct image editing training approaches in **Appendices~A.1** and **Appendices~4.3**.

---

> > ### Author Response · Authors · 2025-11-25
> > **Response to Reviewer K9Zp [Part2]**
> >
> > ## **W2. Concern of strong dependence on image editing quality.**
> >
> > Response: We thank the reviewer for highlighting concerns regarding the reliance on image editing models. To address this, we have added ablation experiments in the revised manuscript using datasets containing partially failed image edits:
> >
> > **[Robustness of the Vision-Zero Framework to Imperfect Image Edits]**
> >
> > To test whether Vision-Zero heavily depends on high-performance image editors, we introduced a 20% noise factor into the RealWorld image dataset to simulate partial editing failures. Specifically, we replaced 10% of the modified images (those provided to the spy) with blank images and another 10% with images identical to the original (civilian) images. We subsequently trained the model on this noisy dataset and evaluated its performance on the validation set.
> >
> > The experimental results, shown in the table below, demonstrate that Vision-Zero continues to improve model performance despite partially failed edits. Although slightly lower than the model trained without noise, the model still achieved an average accuracy improvement of 2.8% compared to the baseline. This indicates that Vision-Zero exhibits substantial robustness to image-editing failures, enabling performance improvements without relying significantly on high-performance image editors.
> >
> > | Model                                     | Mathvista | MathVison | WeMath | MathVerse | LogicVista | DynaMath | Avg. |
> > | ----------------------------------------- | --------- | --------- | ------ | --------- | ---------- | -------- | ---- |
> > | Qwen2.5VL-7b                              | 68.2      | 25.4      | 36.1   | 49.0      | 47.2       | 20.9     | 41.1 |
> > | **VisionZero-Qwen-7B (Real-World)**       | 72.4      | 28.0      | 39.5   | 52.2      | 50.3       | 22.1     | 44.1 |
> > | **VisionZero-Qwen-7B (Real-World+Noise)** | 72.2      | 27.5      | 39.2   | 52.4      | 50.1       | 21.9     | 43.9 |
> >
> > Additionally, we provide examples of model responses to failed image edits in Appendix 4.6, illustrating that the model remains capable of effective reasoning, attempting to leverage alternative information sources to maximize rewards. Consequently, image pairs resulting from unsuccessful edits still serve as valuable training examples.
> >
> > We have detailed the aforementioned experiments and thoroughly discussed Vision-Zero’s robustness to editor performance in **Appendix~4.6** of the revised manuscript.

---

> > > ### Author Response · Authors · 2025-11-25
> > > **Response to Reviewer K9Zp [Part3]**
> > >
> > > ## **W3. Improvement of VisonZero-Qwen-7B(chart) in Chart/OCR is smaller**
> > >
> > > Response: We appreciate the reviewer raising this concern and are happy to provide a detailed explanation.
> > >
> > > **[Why does VisionZero-Qwen-7B (chart) achieve significant improvements on reasoning benchmarks?]**
> > >
> > > Firstly, we clarify why Vision-Zero demonstrates substantial performance improvements on datasets listed in Table 1 after training on chart data. The Vision-Zero framework inherently trains general reasoning capabilities, such as planning and retrieval, through its strategic game environment. The datasets in Table 1, such as MathVision and MathVista, specifically assess general reasoning skills. Hence, regardless of the image distribution used during training, Vision-Zero achieves substantial performance improvements on these reasoning-focused datasets.
> > >
> > >
> > >
> > > **[Why are the numerical improvements smaller on chart/OCR evaluation benchmarks?]**
> > >
> > > Upon careful analysis of the benchmarks used (AI2D, ChartQA, OCR Bench, SEED-2) and the model behavior during evaluation, we identified two primary reasons for the limited numerical improvements:
> > >
> > > - Firstly, there is a **behavioral mismatch between training and evaluation scenarios**. Specifically, during Vision-Zero training, the model is required to reason extensively before responding. In contrast, traditional benchmarks focusing on recognition tasks typically ask for direct answers without intermediate reasoning. This discrepancy makes it difficult for models trained with reasoning strategies to achieve significant improvements or may even lead to performance degradation.
> > >
> > > - Secondly, **these benchmarks exhibit performance saturation**. For instance, Qwen2.5-VL-7B already achieves an accuracy of 86.1% on ChartQA. Even models explicitly trained on chart datasets yield minimal additional improvements (e.g., CHART-R1 using RL on ChartQA only achieves a 2% gain on Qwen2.5-VL-7B; ChartM3, trained via supervised fine-tuning on a 142k chart dataset, only slightly improves Qwen2.5-VL-3B's performance from 83.92% to 84.12%). Therefore, although VisionZero-Qwen-7B (chart) only demonstrates a 1.1% improvement on ChartQA, this modest gain still represents meaningful progress.
> > >
> > >
> > >
> > > **[Experiment: Performance on Reasoning-Oriented Chart Understanding Benchmarks.]**
> > >
> > > To comprehensively validate whether Vision-Zero enhances chart understanding capabilities, we reviewed additional relevant chart understanding research and selected representative benchmarks, including ChartXiv Reasoning, FunctionQA, PaperQA, and ReachQA. These benchmarks require reasoning prior to answering, aligning with the model's training strategy and presenting more challenging tasks. Experimental results, presented in the table below, clearly indicate substantial improvements: VisionZero-Qwen-7B (chart) achieves an average accuracy increase of 3.9% across these four benchmarks. VisionZero-Qwen-7B (clever) and VisionZero-Qwen-7B (realworld) also show varying degrees of performance enhancement due to improved reasoning capabilities, demonstrating Vision-Zero’s effectiveness in chart-related tasks.
> > >
> > > | Model                               | ChartXiV_RQ | FunctionQA | PaperQA  | ReachQA  |
> > > | ----------------------------------- | ----------- | ---------- | -------- | -------- |
> > > | Qwen2.5-VL-7B                       | 42.5        | 82.3       | 68.4     | 50.8     |
> > > | R1-OneVision-7B                     | 35.3        | 69.4       | 64.2     | 46.5     |
> > > | MM-Eureka-Qwen-7B                   | 41.2        | 79.0       | 73.7     | 51.3     |
> > > | VLAA-Thinker-7B                     | 41.3        | 79.0       | 68.4     | 50.4     |
> > > | OpenVLThinker-7B                    | 44.1        | 81.8       | 73.7     | 51.5     |
> > > | ViGaL_Snake+Rotation                | 41.8        | 82.3       | 73.7     | 51.8     |
> > > | **VisionZero-Qwen-7B (CLEVR)**      | 44.3        | 83.3       | 68.4     | 52.0     |
> > > | **VisionZero-Qwen-7B (Real-World)** | 45.2        | 83.8       | 73.7     | 52.5     |
> > > | **VisionZero-Qwen-7B (Chart)**      | **46.6**    | **85.5**   | **73.7** | **53.8** |
> > >
> > > In the revised manuscript, we have included detailed discussions of these experimental results in **Section 3.1**, thoroughly and fairly illustrating Vision-Zero's superior performance in chart understanding tasks.

---

> > > > ### Author Response · Authors · 2025-11-25
> > > > **Response to Reviewer K9Zp [Part4]**
> > > >
> > > > ## **W4. Clarify fine-tuning strategy for understanding the efficiency and scalability of the proposed method.**
> > > >
> > > > **Response:** We appreciate the reviewer raising the issue regarding the ambiguity of our fine-tuning strategy.
> > > >
> > > > **[Clarification on Full-Parameter Fine-Tuning.]**
> > > >
> > > > We would like to clarify that we exclusively employed full-parameter fine-tuning for all training procedures and did not use methods such as LoRA. We have explicitly emphasized this approach in Section 3 of our revised manuscript to avoid any confusion.
> > > >
> > > >
> > > >
> > > > **[Additional Experiments on Vision-Zero’s Training Efficiency.]**
> > > >
> > > >  We have also included more detailed metrics on training costs compared to baseline methods. As demonstrated, Vision-Zero maintains significant advantages in both sample efficiency and total training time relative to various state-of-the-art baselines. It achieves higher model performance within comparable or even shorter training durations.
> > > >
> > > > | Method                         | Num (RL) | Label Cost (Tokens) | Training Method              | Interact | Time Cost          | MMMU     | MMMU_pro |
> > > > | ------------------------------ | -------- | ------------------- | ---------------------------- | -------- | ------------------ | -------- | -------- |
> > > > | Qwen2.5-VL-7B                  | –        | –                   | –                            | –        | –                  | 54.3     | 37.0     |
> > > > | R1-OneVision-7B                | 10k      | > 1.1M              | SFT+GRPO                     | ✗        | ≥ 170 A100-Hours   | 51.9     | 32.6     |
> > > > | VLAA-Thinker-7B                | 25k      | 29.6M               | SFT+GRPO                     | ✗        | ≥ 120 A100-Hours   | 48.2     | 31.9     |
> > > > | OpenVLThinker-7B               | 10k      | 5.7M                | SFT+GRPO                     | ✗        | ≥ 125 A100-Hours   | 54.8     | 22.1     |
> > > > | MM-Eureka-Qwen-7B              | 15k      | –                   | GRPO                         | ✗        | ≈ 700 A100-Hours   | 55.8     | 36.9     |
> > > > | ViGaL-Snake                    | –        | 0                   | RLOO                         | ✗        | ≈ 170 A100-Hours   | 55.8     | 36.6     |
> > > > | **VisionZero-Qwen-7B (CLEVR)** | **2k**   | **0**               | Alternating Self-play + GRPO | **✔**    | **127 A100-Hours** | **58.8** | **37.7** |
> > > >
> > > >
> > > >
> > > > The revised version contains these additional experimental results, detailed descriptions, and further experiments evaluating Vision-Zero’s training efficiency in **Section 3.1**.
> > > >
> > > >
> > > > ## **W5. Limited generalization of image-editing-based augmentation.**
> > > >
> > > > Response: We thank the reviewer for raising concerns about the possibility that Vision-Zero’s advantages primarily arise from image editing-based data augmentation and may not generalize well to other tasks.
> > > >
> > > > **[Vision-Zero Does Not Solely Depend on Data Augmentation Effects.]**
> > > >
> > > > As addressed in our response to Weakness 1, Vision-Zero's performance improvements are not merely due to data augmentation. Instead, the strategic game environment and self-play framework foster enhanced reasoning capabilities within the model. Experimental comparisons with purely data-augmentation-based methods, such as MiCo, clearly validate this point, showing significantly higher performance gains achieved by the Vision-Zero framework.
> > > >
> > > > **[Potential Limitations in Generalization and Domain Transfer]**
> > > >
> > > > We appreciate the reviewer’s valuable feedback and have added a detailed discussion of Vision-Zero's potential limitations in **Appendix~A.5** of the revised manuscript.
> > > >
> > > > Specifically, first, due to its reliance on differentiated image pairs, implementing Vision-Zero requires the use of image editors, which might present limitations in highly specialized or resource-constrained domains (e.g., medical imaging, scientific charts, remote sensing), where such edited data may not always be readily available. Second, the current framework is based on single-image observations and pairwise edits; thus, extending it to richer modalities, such as extended videos, complex multi-image contexts, or interactive 3D environments, may necessitate substantial redesign of the game mechanisms and training algorithms. Addressing these limitations is an important direction for future research.

---

> > > > > ### Author Response · Authors · 2025-11-25
> > > > > **Response to Reviewer K9Zp [Part5]**
> > > > >
> > > > > We kindly remind that the questiones you raised appears to be unrelated to our paper. As we have already included a number of additional experiments in this rebuttal (as discussed in the general response), we would like to know if there are any other questions you may have, and we are very happy to adress. Thank you again for your thoughtful review and constructive feedback!

---

### Official Review · Reviewer_CCNi · 2025-10-31

**Soundness:** 3
**Presentation:** 2
**Contribution:** 3
**Rating:** 4
**Confidence:** 3

**Summary:**

This paper introduces a zero-human-in-the-loop training framework for VLMs using a gamified self-play mechanism inspired by social deduction games like "Who Is the Spy?", eliminating the need for expensive human-annotated datasets.
The game involves multiple agents (civilians and a spy) who observe slightly different images and must describe or deduce the differences through dialogue. The model improves via Iterative Self-Play Policy Optimization (Iterative-SPO), which alternates between self-play and reinforcement learning with verifiable rewards (RLVR) to avoid performance plateaus.

**Strengths:**

1. First to apply self-play to VLMs in a gamified adversarial setting.
2. Propose a novel training algorithm that avoids equilibrium stagnation by alternating between self-play and RLVR.
3. The evaluated domains are quite comprehensive, including math, charts, vision-centric, and simulated images.

**Weaknesses:**

- The improvement of the Chart/OCR is marginal, as shown in Tab.2.
- The data preparation cost in the Tab.3 is ambiguous, where the statistics might be further explained.
- The improvement over the InternVL series is marginal compared to the QwenVL series, as shown in Tab. 4, casting doubts on its generability.
- The intuition/theory between the "who is a spy" proxy task and the general question-answering tasks remains unclear.

**Questions:**

- Will the models cheat or find shortcuts via editing methods? Because the editing possibilities/categories are pre-defined. (e.g., swapping numerical attributes in chart data.)
- Is this idea novel in the multi-agent community? I have noticed that Role-Advantage Estimation is based on the previous research. What's your differences?

---

> ### Author Response · Authors · 2025-11-24
> **Response to Reviewer CCNi [Part1]**
>
> We would like to sincerely thank reviewer CCNi for taking the time to review our paper and provide valuable feedback. We appreciate the opportunity to address the questions and concerns, and we believe the revisions will further strengthen our work.
>
>
>
> ## **W1. The improvement of the Chart/OCR is marginal.**
>
> **Response:** We thank the reviewer for pointing this out and are happy to provide a more detailed explanation.
>
> **[Why do the improvements on Chart/OCR benchmarks appear small?]**
>
> After carefully analyzing the benchmarks we used (AI2D, ChartQA, OCRBench, SEED-2) and inspecting model behavior under the VLMEvalKit evaluation protocol, as well as reviewing related work, we found two main reasons why the numerical gains in Tab.~2 look modest:
>
> - Firstly, these benchmarks primarily assess relatively straightforward recognition abilities, resulting in **near-saturated model performance**. For example, Qwen2.5-VL-7B already achieves 86.1% accuracy on ChartQA, and even methods specifically designed for chart understanding obtain only modest absolute gains on it: Chart-R1[1] performs RL directly on ChartQA and improves Qwen2.5-VL-7B by about 2% on the test set, while ChartM3[2], after supervised fine-tuning on 142k chart samples, improves Qwen2.5-VL-3B from 83.92% to 84.12%. Therefore, the 1.1% gain of VisionZero-Qwen-7B(Chart) on ChartQA is actually comparable to the improvements reported by highly specialized chart models.
> - Secondly, there is a **behavioral discrepancy between model training and evaluation.** During Vision-Zero training, the model is explicitly encouraged to “think before answering”. In contrast, standard Chart/OCR benchmarks in VLMEvalKit evaluate models in a pure “direct answer” mode, the prompt is like "Please give the answer directly". This mismatch in behavior makes it inherently difficult for a reasoning-focused training scheme to translate into large improvements on these benchmarks.
>
>
>
> **[Experiment 1: Results on Reasoning-based Chart Understanding Benchmarks.]**
>
> To more directly evaluate whether Vision-Zero improves chart understanding, we additionally consider chart benchmarks that require explicit reasoning and are more aligned with our training setting: ChartXiv_Reasoning, FunctionQA, PaperQA, and ReachQA. These tasks explicitly prompt models to reason before answering and are substantially more challenging. As shown in Table below, VisionZero-Qwen-7B(Chart) achieves significantly larger gains on these benchmarks, improving accuracy by an average of +3.9% across the four datasets.
>
> | Model                               | ChartXiV_RQ | FunctionQA | PaperQA  | ReachQA  |
> | ----------------------------------- | ----------- | ---------- | -------- | -------- |
> | Qwen2.5-VL-7B                       | 42.5        | 82.3       | 68.4     | 50.8     |
> | R1-OneVision-7B                     | 35.3        | 69.4       | 64.2     | 46.5     |
> | MM-Eureka-Qwen-7B                   | 41.2        | 79.0       | 73.7     | 51.3     |
> | VLAA-Thinker-7B                     | 41.3        | 79.0       | 68.4     | 50.4     |
> | OpenVLThinker-7B                    | 44.1        | 81.8       | 73.7     | 51.5     |
> | ViGaL_Snake+Rotation                | 41.8        | 82.3       | 73.7     | 51.8     |
> | **VisionZero-Qwen-7B (CLEVR)**      | 44.3        | 83.3       | 68.4     | 52.0     |
> | **VisionZero-Qwen-7B (Real-World)** | 45.2        | 83.8       | 73.7     | 52.5     |
> | **VisionZero-Qwen-7B (Chart)**      | **46.6**    | **85.5**   | **73.7** | **53.8** |
>
>
>
> **[Experiment 2: Comparison with chart-specialized models.]**
>
> We further compare Vision-Zero with models that are explicitly post-trained for chart understanding, including ECD [3], Bespoke-MiniChart-7B [4], and Chart-R1-7B [1]. As summarized in Table below, VisionZero-Qwen-7B(Chart) trained without any QA annotations but achieves performance that is comparable to or even better than models trained on large-scale chart QA data. This indicates that our strategic game environment enables strong chart understanding.
>
> | Model                          | Chart Number | QA Number | ChartXiV_RQ | ReachQA  | Avg.     |
> | ------------------------------ | ------------ | --------- | ----------- | -------- | -------- |
> | Qwen2.5VL-7b                   | -            | -         | 42.5        | 50.8     | 46.7     |
> | ECD-Qwen2.5VL-7b               | 10.5k        | 320k      | 40.2        | 53.5     | 46.9     |
> | Bespoke-MiniChart              | 13.0k        | 91k       | 46.2        | **54.0** | 50.1     |
> | Chart-R1-7B                    | 21.9k        | 32.7k     | 46.2        | -        | -        |
> | **VisionZero-Qwen-7B (chart)** | **2.0k**     | **0**     | **46.6**    | 53.8     | **50.2** |
>
> In the revised version, we have comprehensively reported these experimental findings in **Section 3.1** and the **Appendix~4.4** to thoroughly and fairly illustrate Vision-Zero's outstanding performance on chart understanding tasks.

---

> ### Author Response · Authors · 2025-11-24
> **Response to Reviewer CCNi [Part2]**
>
> ## **W2. The data preparation cost in the Tab.3 is ambiguous.**
>
> **Response:** We thank the reviewer for the valuable suggestion. In response, we have revised Table 3 in the main paper accordingly.
>
> **【Experiment: Comparison of Data Construction and Training Costs Between Vision-Zero and SOTA Baselines.】**
>
>  As shown in Table below, we have added more detailed metrics regarding data preparation and training costs. The "label cost" represents the number of tokens generated by teacher or judging LLMs during data curation, with all token counts recalculated using the Qwen2.5 tokenizer for consistency. Methods such as VIGAL and Vision-Zero, which utilize unlabeled data, incur no label costs. For estimating the training time cost, we referred to the paper of the baseline methods to identify the number of training samples used during RL training, multiplying this number by a standardized GRPO sample cost to approximate the training time under an equitable setting. The notation "≥" for some models indicates that only RL costs were calculated, excluding SFT costs. The training time for Vision-Zero is based on our measured results.
>
> | Method                         | Num (RL) | Label Cost (Tokens) | Training Method              | Interact | Time Cost          | MMMU     | MMMU_pro |
> | ------------------------------ | -------- | ------------------- | ---------------------------- | -------- | ------------------ | -------- | -------- |
> | Qwen2.5-VL-7B                  | –        | –                   | –                            | –        | –                  | 54.3     | 37.0     |
> | R1-OneVision-7B                | 10k      | > 1.1M              | SFT+GRPO                     | ✗        | ≥ 170 A100-Hours   | 51.9     | 32.6     |
> | VLAA-Thinker-7B                | 25k      | 29.6M               | SFT+GRPO                     | ✗        | ≥ 120 A100-Hours   | 48.2     | 31.9     |
> | OpenVLThinker-7B               | 10k      | 5.7M                | SFT+GRPO                     | ✗        | ≥ 125 A100-Hours   | 54.8     | 22.1     |
> | MM-Eureka-Qwen-7B              | 15k      | –                   | GRPO                         | ✗        | ≈ 700 A100-Hours   | 55.8     | 36.9     |
> | ViGaL-Snake                    | –        | 0                   | RLOO                         | ✗        | ≈ 170 A100-Hours   | 55.8     | 36.6     |
> | **VisionZero-Qwen-7B (CLEVR)** | **2k**   | **0**               | Alternating Self-play + GRPO | **✔**    | **127 A100-Hours** | **58.8** | **37.7** |
>
> Our findings clearly demonstrate Vision-Zero's significant advantages in sample efficiency and total training time compared to various SOTA baselines, achieving superior model performance within comparable or even shorter training durations. These benefits primarily stem from Vision-Zero's challenging interactive training environment.
>
> We have updated **Table 3** in the revised version to include these results and provided a detailed explanation of our cost estimation methodology in **Appendix~3.3**.

---

> ### Author Response · Authors · 2025-11-24
> **Response to Reviewer CCNi [Part3]**
>
> ## **W3. Improvement over the InternVL is marginal compared to the QwenVL.**
>
> **Response:** We appreciate the reviewer’s concern about the gains on the InternVL series and the question of generability.
>
> **[Why are the gains on InternVL smaller than on QwenVL?]**
>
> In fact, the difference in improvement magnitude between the QwenVL and InternVL series is **fully expected**. InternVL3‑8B/14B has already undergone extensive optimization for mathematical and reasoning tasks and achieves substantially higher baseline performance than Qwen2.5‑VL‑7B on these benchmarks [5]. As a result, its remaining headroom for further improvement is inherently smaller, and any post-training method can naturally provide only limited absolute gains.
>
> Under such saturation, the average improvements of **+1.8%** and **+1.6%** accuracy that Vision-Zero brings over InternVL3‑8B and InternVL3‑14B, respectively, are both **reasonable and meaningful.** For comparison, classical VLM reasoning models such as SCS [6], which is trained on large amounts of labeled data, report average gains of only around 1.6% on InternVL3-8B across VLM reasoning benchmarks.
>
>
>
> **[Experiment: Vision-Zero vs. GRPO on InternVL models.]**
>
> To further demonstrate that the gains on InternVL are significant and non-trivial, we train InternVL3-8B and InternVL3-14B with vanilla GRPO on the MM-Eureka training set under the same setting as Vision-Zero, and evaluate all models on benchmarks. As shown in the Table below, both VisionZero-InternVL3-8B and VisionZero-InternVL3-14B clearly outperform the MM-Eureka (GRPO) baselines on most benchmarks, confirming that Vision-Zero brings additional benefits beyond standard GRPO even for these strong backbones.
>
>
>
> | Model                       | MathVista | MathVision | WeMath   | MathVerse | LogicVista | DynaMath | Avg.     |
> | --------------------------- | --------- | ---------- | -------- | --------- | ---------- | -------- | -------- |
> | InternVL3-8B                | 60.4      | 21.3       | 26.8     | 32.2      | 40.5       | 26.8     | 34.7     |
> | MM-Eureka-InternVL-8B       | 62.4      | 22.1       | 26.8     | 32.1      | 38.9       | 28.7     | 35.2     |
> | **VisionZero-InternVL3-8B** | **62.2**  | **24.2**   | **28.7** | **32.9**  | **41.8**   | **29.2** | **36.5** |
>
>
>
> | Model                        | MathVista | MathVision | WeMath   | MathVerse | LogicVista | DynaMath | Avg.     |
> | ---------------------------- | --------- | ---------- | -------- | --------- | ---------- | -------- | -------- |
> | InternVL3-14B                | 74.1      | 33.8       | 42.3     | 43.3      | 51.6       | 30.1     | 45.8     |
> | MM-Eureka-InternVL-14B       | 75.2      | 34.5       | 42.5     | 44.2      | 52.0       | 30.9     | 45.4     |
> | **VisionZero-InternVL3-14B** | **75.4**  | **34.8**   | **44.9** | **45.1**  | **53.1**   | **31.3** | **47.4** |
>
> We also present the training efficiency comparison in Figure 7(b) of the revised paper, where Vision‑Zero achieves **6.4× higher sample efficiency** than GRPO on InternVL3‑8B, further demonstrating its strong effectiveness on the InternVL family.
>
> Finally, we emphasize that **Qwen2.5‑VL‑7B and InternVL3 differ significantly** in visual encoders, pre-training pipelines, and training recipes. Yet, Vision‑Zero consistently improves both model families, across **multiple parameter scales (7B, 8B, 14B)**. We believe this cross-architecture, cross-scale effectiveness is strong evidence of the generability of Vision-Zero.
>
> We have added these results and the discussion on the generality of Vision-Zero to **Table 4** and **Section 3.2** in the revised version.
>
> [1] Chart-R1: Chain-of-Thought Supervision and Reinforcement for Advanced Chart Reasoner.
>
> [2] ChartM3: A Multi-Stage Code-Driven Pipeline for Constructing Multi-Dimensional and Multi-Step Visual Reasoning Data in Chart Comprehension.
>
> [3] Effective Training Data Synthesis for Improving MLLM Chart Understanding.
>
> [4] Bespoke-MiniChart-7B: Pushing The Frontiers Of Open VLMs For Chart Understanding.
>
> [5] InternVL3: Exploring Advanced Training and Test-Time Recipes for Open-Source Multimodal Models.
>
> [6] Enhancing the Outcome Reward-based RL Training of MLLMs with Self-Consistency Sampling.

---

> ### Author Response · Authors · 2025-11-24
> **Response to Reviewer CCNi [Part4]**
>
> ## **W4. Unclear intuition/theory between "who is a spy" and general QA task .**
>
> **Response:** We appreciate the reviewer’s concern and agree that the connection between our game and general QA tasks should be made more explicit.
>
> **[Why does the “Who is the Spy?” game improve general QA performance?]**
>
> The key reason is that **the skills required to play the “Who is the Spy?” are highly aligned with the skills needed for general multimodal QA.**  For instance, during the clue phase, the model must identify differences between images and textual descriptions, directly enhancing fine-grained visual comparison and understanding abilities. Additionally, the necessity to integrate and avoid contradictions or repetitions during multiple clue exchanges develops the model’s multi-step reasoning and context integration skills. These enhanced abilities directly translate into improved performance on general tasks.
>
> Furthermore, the strategic and challenging nature of the "Who is Spy" game environment **encourages meticulous reasoning and logical thinking in model responses**, beneficial to general QA tasks. As depicted in Fig 5 of the main text, the trained model exhibits superior capabilities in planning, retrieval, decomposition, strategizing, and logical reasoning, all critical skills for answering general QA tasks.
>
>
>
> **[Experiment: Comparison of CoTs before and after training.]**
>
> To illustrate how playing the "Who is Spy" game enhances model performance in general QA tasks more intuitively, we present examples of CoTs before and after training on general QA tasks. As illustrated in the following block (the fig is in Appendix 4.5 Example1 in the revised paper), the comparison clearly demonstrates that the trained model displays more explicit thinking strategies, while also more accurately focusing on key image information. This directly results in more accurate model responses.
>
> ```prompt
> (Fig is in Appendix A.4.5 Example1)
> Question: How many countries have people working for more than 35 hours?
>
> Untrained model:
> The graph compares the average usual weekly hours worked by women over the age of 15 across three countries (Romania, Portugal, and Switzerland) from 1995 to 2007. Each country is represented by a different colored line: Romania is blue, Portugal is orange, and Switzerland is dark blue. From 1995 to 2007, the line representing Romania (blue) is consistently above the 35-hour mark on the y-axis, which measures average usual weekly hours. The lines for Portugal and Switzerland are below the 35-hour mark throughout this period. Thus, only one country, Romania, shows average weekly hours worked exceeding 35 hours over the entire period displayed. \boxed{1}
>
> Trained model:
> To determine how many countries have people working for more than 35 hours on average weekly over the years from 1995 to 2007, let's analyze the data shown in the graph step by step:
> 1. **Identify the countries represented:**
>    - Romania (in blue)
>    - Portugal (in orange)
>    - Switzerland (in light blue)
> 2. **Analyze the hours worked for each country:**
>    - **Romania:** The blue line representing Romania shows that the average usual weekly hours worked remain consistently above 35 hours throughout the entire period from 1995 to 2007.
>    - **Portugal:** The orange line representing Portugal shows that the average usual weekly hours worked also remain consistently above 35 hours over the same period.
>    - **Switzerland:** The light blue line representing Switzerland shows that the average usual weekly hours worked remain consistently below 35 hours over the entire period.
> 3. **Determine the countries with averages above 35 hours:**
>    - From the above analysis, Romania and Portugal have average usual weekly hours worked consistently above 35 hours.
>    - Switzerland has average usual weekly hours worked consistently below 35 hours.
>
> Therefore, the number of countries with people working for more than 35 hours on average weekly over the years is **2**.
> \boxed{2}
> ```
>
>
>
> In the revised version, we have included additional examples in the **Appendix~4.5** to demonstrate why Vision-Zero enhances performance on general tasks and have further discussed the connections between the "Who is Spy" game and general tasks.

---

> ### Author Response · Authors · 2025-11-25
> **Response to Reviewer CCNi [Part5]**
>
> ## **Q1.Will the models cheat or find shortcuts via editing methods?**
>
> **Response:** We appreciate the reviewer’s concern regarding potential shortcuts arising from pre-defined editing operators. Actually, during training, it is hard for the model to cheat or find shortcuts through editing methods because of the following two points.
>
> Firstly, **our editing targets are randomly generated**. For instance, in CLEVR Dataset, edited objects vary randomly in position, material, and size, leaving no fixed spatial or visual patterns that could allow shortcut learning. Similarly, in the Chart Dataset, different series, bars, or segments are randomly selected for value swapping or perturbation.
>
> Secondly, **our edits leave no detectable artifacts**; for example, we re-render entirely new images for CLEVR Dataset, and redraw charts using Python scripts for Chart Dataset. The model never sees an editing mask, and there are no systematic graphical glitches that could be used as a cheap cue for “this image was edited.”
>
> Our experimental results further support this claim. As shown in Figures 6(b) and (c) of the main text, the lengths of tokens generated during both the clue and decision phases gradually increase throughout training. This indicates that the model does not find shortcuts or cheats, which would otherwise result in shorter and quicker responses.
>
>
>
> ## **Q2. Is this idea novel in the multi-agent community?**
>
> **Response:** We thank the reviewer for raising this question and highlighting the connection to prior multi-agent work.
>
> **[Clarification Regarding Role-Advantage Estimation (RAE)]**
>
> Firstly, we wish to clarify that we do not claim Role-Advantage Estimation (RAE) as a novel algorithmic contribution or as part of our innovation. Our RAE component follows the design proposed by SPIRAL [6] to enhance training stability under conditions of asymmetric role information. In the revised version, we explicitly state that in order to avoid any misunderstanding.
>
> **[Novelty of Vision-Zero relative to the multi-agent community.]**
>
> We fully understand the reviewer’s concern that there exist multi-agent works that also use social deduction games similar to “Who is the Spy?”[7] [8] [9]. Our key contributions differ from these works along several axes:
>
> First, to our knowledge, **Vision-Zero is the first to formulate a social deduction game in a** **self-play framework** **specifically for post-training VLMs**. Existing works on Undercover-style games typically use LLM/VLM agents with fixed parameters and focus on measuring or improving in-game social behavior via prompt engineering and interaction design. In contrast, Vision-Zero performs full-parameter fine-tuning of a shared VLM policy through self-play in the “Who is the Spy?” environment, and we show that this training leads to consistent improvements on diverse downstream benchmarks.
>
> Secondly, to ensure stable model training, **Vision-Zero introduces the Iterative-SPO training algorithm.** As the first effort to train VLM in social reasoning games, we identified the instability caused by joint training across multiple stages of gameplay. Consequently, we introduced an Iterative Self-Play Policy Optimization (Iterative-SPO) algorithm that alternates between (i) a zero-sum clue stage with role-conditioned baselines and KL regularization, and (ii) a decision stage trained with RLVR and group-normalized advantages. This multi-stage, alternating optimization scheme is tailored to the VLM social-game setting and, to our knowledge, has not been explored in prior multi-agent work.
>
> Thirdly, **Vision-Zero explicitly targets general reasoning capability.** Prior research in the multi-agent domain primarily aimed at optimizing gameplay or inter-model collaboration, without addressing the broader improvement and transferability of model capabilities. By cleverly incorporating diverse unlabeled image pairs, Vision-Zero expands game inputs from a singular distribution to arbitrary distributions, significantly boosting model reasoning performance on general tasks.
>
> We further emphasize the differences and novelty of vision-zero compared to these works in the **Appendix~A.1** of the revised version.
>
> [6] SPIRAL: Self-Play on Zero-Sum Games Incentivizes Reasoning via Multi-Agent Multi-Turn Reinforcement Learning.
>
> [7] Who is Undercover? Guiding LLMs to Explore Multi-Perspective Team Tactic in the Game.
>
> [8] Guiding llms to explore multi-perspective team tactic in the game.
>
> [9] Leveraging word guessing games to assess the intelligence of large language models.

---

### Official Review · Reviewer_gbyd · 2025-11-01

**Soundness:** 3
**Presentation:** 4
**Contribution:** 3
**Rating:** 8
**Confidence:** 3

**Summary:**

The paper presents VISION-ZERO, a framework that enhances the reasoning capabilities of vision–language models (VLMs) through self-play and RLVR, thereby reducing reliance on costly human-annotated data. Specifically, the framework constructs a strategic environment inspired by the game “Who Is the Spy”, in which training data are naturally generated through self-play. The resulting data are then used to iteratively optimize VLMs using the Iterative-SPO algorithm. Extensive empirical results on multiple benchmarks demonstrate that VISION-ZERO surpasses baseline models, significantly enhancing the reasoning ability of VLMs while avoiding cross-capability negative transfer and substantially reducing dataset construction costs.

**Strengths:**

1. This paper is well-motivated. Existing RLVR methods rely heavily on carefully constructed datasets. By introducing self-play, this method enables reasoning improvement on label-free data, helping to extend RLVR to broader domains where manual annotation is difficult or impractical.
2. The experiments are comprehensive. Evaluations on **Task Generalization Capability**, **Cross-Capability Negative Transfer Mitigation**, and **Low Dataset Construction Costs** strongly demonstrate the superiority of **VISION-ZERO**, providing solid evidence for its effectiveness in mitigating data scarcity and improving scalability.
3. The framework proposed in this paper is novel. The *“Who Is the Spy”* game environment appears to be highly scalable and can be readily extended to diverse domains. This offers a new paradigm for enhancing the capabilities of vision–language models (VLMs).

**Weaknesses:**

1. The data curation pipeline is primarily focused on the **CLEVR** and **Chart** domains. As mentioned in *Appendix 2.2*, directly editing chart images using **NanoBanna** and **ChatGPT** can be extremely challenging. This raises concerns about the scalability of the proposed framework, particularly regarding how to generate image pairs with arbitrary data.
2. The overall design of this framework appears to be sophisticated. The success of the framework relies on the effective coordination among **self-play in the game environment**, **Self-Play Optimization in the Clue Stage**, and **RLVR in the Decision Stage**. A deeper discussion on the stability of this framework would be valuable for understanding its robustness and generalization.

**Questions:**

1. Ablation studies on critical components and hyperparameters (e.g., RAE and the number of civilians) would be helpful for further understanding the contribution of each module and the sensitivity of the framework.
2. If **VISION-ZERO** is trained on static datasets, is there a possibility of **knowledge leakage** when the model encounters the same image pairs during training?

**Details Of Ethics Concerns:**

no concerns

---

> ### Author Response · Authors · 2025-11-24
> **Response to Reviewer gbyd [Part1]**
>
> We would like to thank the reviewer for taking the time to review our paper and provide valuable feedback. We are glad that the reviewer thinks our paper is well-motivated and novel. We are excited to have the chance to address the reviewer’s questions and concerns. These edits will make the paper stronger.
>
>
>
> ## **W1. Concerns about the scalability because of the limited edit ability.**
>
> **Response:** We thank the reviewer for raising concerns regarding scalability and reliance on editing-based data curation. We agree that directly editing images, such as charts using editors like Nano Banana or ChatGPT, may pose practical challenges. To address this concern, we provide responses from two perspectives:
>
> **[Vision-Zero is robust to imperfect or failed edits and does not strictly rely on strong editors.]**
>
> To test whether Vision-Zero critically depends on a highly reliable image editor, we simulate partial editing failures on the real-world subset. Specifically, we inject 20% noisy pairs into the training data: we replace 10% of the modified images (those shown to the spy) with blank images, and another 10% with exact copies of the original images (i.e., no actual edit). We then train Vision-Zero on this noisy dataset under the same setting and evaluate the resulting model on the validation benchmarks.
>
> The results, presented in the table below, demonstrate that even with partially failed edits, Vision-Zero still significantly improved model performance. Although slightly lower than models trained without noise, there was an average accuracy improvement of 2.8% compared to the original model. This indicates that Vision-Zero exhibits strong robustness to editing failures and can enhance model performance without relying heavily on highly capable image editors.
>
> | Model                                     | Mathvista | MathVison | WeMath | MathVerse | LogicVista | DynaMath | Avg. |
> | ----------------------------------------- | --------- | --------- | ------ | --------- | ---------- | -------- | ---- |
> | Qwen2.5VL-7b                              | 68.2      | 25.4      | 36.1   | 49.0      | 47.2       | 20.9     | 41.1 |
> | **VisionZero-Qwen-7B (Real-World)**       | 72.4      | 28.0      | 39.5   | 52.2      | 50.3       | 22.1     | 44.1 |
> | **VisionZero-Qwen-7B (Real-World+Noise)** | 72.2      | 27.5      | 39.2   | 52.4      | 50.1       | 21.9     | 43.9 |
>
> Additionally, in Appendix 4.6, we included examples of model responses when encountering failed edited image pairs. It is observed that even with imperfect edits, the model still engages in reasoning and strives to utilize alternative information to achieve high rewards. Thus, imperfectly edited image pairs remain effective training samples.
>
>
>  **[Improving image editors will further lower the cost of constructing diverse image pairs.]**
>
> Finally, we note that image editing models are advancing rapidly, and recent tools like Gemini-Pro and GPT-5.1 already support more reliable, fine-grained edits on complex images. As these tools continue to improve, we expect the cost of generating diverse edited image pairs in new domains to further decrease, making it easier to instantiate Vision-Zero in arbitrary settings.
>
> We have added the noisy-edit experiment and a detailed discussion of Vision-Zero’s robustness to editor quality to **Appendix~4.6** in the revised manuscript to better address the reviewer’s concern.
>
>
> ## **Q2:  Possibility of knowledge leakage during training.**
>
>
>
> **Response:** We thank the reviewer for raising this question regarding potential knowledge leakage from training on a static set of image pairs. In practice, due to Vision-Zero’s low dataset construction costs, **it does not rely heavily on repetitive use of small datasets**. In actual training practice, each image pair is sampled only a limited number of times throughout the entire training process. Our experimental results demonstrating consistent performance improvements across various benchmarks further support this point.

---

> > ### Author Response · Authors · 2025-11-24
> > **Response to Reviewer gbyd [Part2]**
> >
> > ## **W2&Q1. More ablation study of paramters and critical components.**
> > **Response:** We thank the reviewer for emphasizing the importance of understanding both the stability of our multi-stage framework and the contributions of its key components. In response, we have added three sets of experiments in the revised paper to explicitly illustrate and discuss the stability and dependency of Vision-Zero on its core components. All experiments were conducted using the CLEVR dataset and the Qwen2.5-VL-7B model, with uniform training for 100 iterations.
> >
> > **[Experiment 1: Ablation on the Number of Players]**
> >
> > We first evaluated whether Vision-Zero could consistently enhance model performance across different player counts. We trained models with varying numbers of civilian players (2, 3, and 4), keeping the spy count constant at one. As illustrated in the table below, Vision-Zero consistently improved model performance across different player counts, demonstrating robustness regarding player numbers. With more players, the game environment becomes more challenging, increasing the reasoning steps and rewards per sample, leading to progressively improved model performance. Even with only two civilians, the model still achieved an average performance improvement of 1.3% across six benchmarks.
> > | Model                                 | Mathvista | MathVison | WeMath | MathVerse | LogicVista | DynaMath | Avg. |
> > | ------------------------------------- | --------- | --------- | ------ | --------- | ---------- | -------- | ---- |
> > | Qwen2.5VL-7b                          | 68.2      | 25.4      | 36.1   | 49.0      | 47.2       | 20.9     | 41.1 |
> > | VisionZero-Qwen-7B (Civ=2) | 69.7      | 26.5      | 37.2   | 50.6      | 48.9       | 21.2     | 42.4 |
> > | VisionZero-Qwen-7B (Civ=3) | 72.6      | 28.1      | 39.8   | 51.9      | 50.1       | 22.3     | 44.1 |
> > | VisionZero-Qwen-7B (Civ=4) | 73.2      | 28.0      | 40.0   | 52.1      | 51.8       | 24.1     | 44.9 |
> >
> > **[Experiment 2: Ablation on the Number of Clue Rounds]**
> >
> > We also investigated Vision-Zero's performance when varying the number of clue rounds per game (1, 2, and 3). Results presented in the table below show minimal performance improvement with only one clue round due to insufficient information for the decision-making stage and fewer information integration during training. However, when the clue rounds increased to two or three, the model exhibited significant performance enhancements, averaging improvements of 3% and 4.1%, respectively, across six benchmarks. This trend indicates that as the clue rounds increase, the model’s capacity for handling and integrating information improves, further enhancing performance.
> > | Model                                        | Mathvista | MathVison | WeMath | MathVerse | LogicVista | DynaMath | Avg. |
> > | -------------------------------------------- | --------- | --------- | ------ | --------- | ---------- | -------- | ---- |
> > | Qwen2.5VL-7b                                 | 68.2      | 25.4      | 36.1   | 49.0      | 47.2       | 20.9     | 41.1 |
> > | VisionZero-Qwen-7B (Clue\_Rd = 1) | 68.0      | 26.5      | 36.5   | 48.2      | 47.2       | 21.2     | 41.3 |
> > | VisionZero-Qwen-7B (Clue\_Rd = 2) | 72.6      | 28.1      | 39.8   | 51.9      | 50.1       | 22.3     | 44.1 |
> > | VisionZero-Qwen-7B (Clue\_Rd = 3) | 73.1      | 29.0      | 40.5   | 52.2      | 52.0       | 24.3     | 45.2 |
> >
> >
> >
> > **[Experiment 3: Ablation on the Role-Advantage Estimation (RAE) Module]**
> >
> > To confirm the significance of the RAE module, we trained models with and without RAE and compared their performances. In the no-RAE scenario, rewards calculated after the clue stage did not deduct the role-advantage baseline but were directly applied for weight updates. Results shown in the table below indicate that omitting the RAE module leads to negative performance outcomes. This is because the spy and civilian roles inherently differ in information access and task complexity. Directly applying rewards without adjusting for role advantages overlooks the actual performance level of the model, impairing correct learning. This clearly demonstrates the critical importance of the RAE module.
> >
> > | Model                                   | Mathvista | MathVison | WeMath | MathVerse | LogicVista | DynaMath | Avg. |
> > | --------------------------------------- | --------- | --------- | ------ | --------- | ---------- | -------- | ---- |
> > | Qwen2.5VL-7b                            | 68.2      | 25.4      | 36.1   | 49.0      | 47.2       | 20.9     | 41.1 |
> > | VisionZero-Qwen-7B (w/ RAE)  | 72.6      | 28.1      | 39.8   | 51.9      | 50.1       | 22.3     | 44.1 |
> > | VisionZero-Qwen-7B (w/o RAE) | 65.2      | 21.3      | 30.1   | 47.2      | 44.3       | 16.1     | 37.4 |
> >
> >
> >
> > In the revised version, we have detailed these experimental results and provided comprehensive analyses discussing Vision-Zero’s stability and robustness in **Appendix~4.7**.

---

### Author Response · Authors · 2025-11-25
**Summary of Revisions and Global Response**

We sincerely thank all reviewers for their insightful comments and valuable suggestions, which have significantly helped us to improve the manuscript.

In this paper, we propose **Vision-Zero**, a novel self-play framework designed to enhance the reasoning capabilities of VLMs. By engaging VLMs in strategic reasoning and actions within a "Who is Spy" game environment and introducing an Iterative Self-Play Policy Optimization algorithm, Vision-Zero notably enhances the general reasoning abilities of VLMs.

We are greatly encouraged by the reviewers’ positive feedback on our motivation and experimental design. Specifically, reviewers highlighted:

- **‘This paper is well-motivated.’**(Reviewer gbyd, Reviewer K9Zp)
- **‘First to apply self-play to VLMs in a gamified adversarial setting.’**(Reviewer CCNi)
- **‘The evaluated domains are quite comprehensive.’** (Reviewer CCNi, Reviewer gbyd, Reviewer B6Um)
- **‘The Spy–Player game formulation is an interesting and original idea that provides a novel self-supervised interaction mechanism.’** (Reviewer K9Zp)
- **‘demonstrates that self-play visual game can be an effective task for VLM post-training, which is an interesting finding.’**(Reviewer B6Um)

Reviewers provided many insightful and constructive suggestions. We have endeavored to address all concerns by adding detailed descriptions and required results. Below is a summary of the primary modifications:

- **Detailed Comparison of Training Efficiency and Cost (Reviewer B6Um, Reviewer CCNi, Reviewer K9Zp):** Following reviewers’ suggestions, we provided a detailed comparative analysis of training efficiency and training costs of Vision-Zero. Refer to Section 3.1 (Tab. 3 & Fig. 7) in the revised version.
- **Stability Experiments under Image Editing Failures (Reviewer gbyd, Reviewer K9Zp):** As suggested, we simulated partial editing failures in the training dataset to test Vision-Zero's robustness under these conditions. Refer to Appendix A.4.6 in the revised version.
- **Comparison with Direct Training on Edited Image Pairs / Contrastive RL (Reviewer B6Um, Reviewer K9Zp):** Following reviewers' recommendations, we compared Vision-Zero with simpler direct training approaches like MiCo under identical training settings. See Appendix A.1 and Appendix A.4.3 in the revised version.
- **Evaluation and Comparison on Reasoning-based Chart Understanding Benchmarks (Reviewer CCNi, Reviewer K9Zp):** To further validate Vision-Zero's performance in the chart understanding domain, we added evaluation results on four more challenging chart understanding benchmarks and compared them with chart-specialized models. Refer to Section 3.1 (Tab. 2) and Appendix A.4.4 in the revised version.
- **Ablation Studies on Parameters and Key Components (Reviewer gbyd):}** As recommended, we conducted ablation studies on critical parameters of Vision-Zero, including the number of players, the number of clue rounds, and the key component RAE. Refer to Appendix A.4.7 in the revised version.
- **Chain-of-Thought Observations on General QA Tasks Pre- and Post-Training (Reviewer CCNi):** We supplemented comparisons of model responses on general QA tasks before and after Vision-Zero training to better understand its effectiveness in enhancing model performance. Refer to Appendix A.4.7 in the revised version.
- **Discussion on Vision-Zero’s Limitations (Reviewer K9Zp):** Following the reviewer's advice, we included a comprehensive discussion on the limitations of Vision-Zero and potential future research directions. Refer to Appendix A.5 in the revised version.

The reviewers' invaluable feedback has significantly improved our paper. We'd be very happy to answer any further questions.

Looking forward to the reviewers' feedback :D

---

### Author Response · Authors · 2025-11-30
**Rebuttal Summary and Acknowledgement**

To the newly assigned Area Chair, Senior Area Chair, Program Chair and Reviewers,

We have received the recent updates regarding the data leakage incident. We deeply regret the disruption caused by this event and fully recognize the immense pressure it has placed on the organizers and volunteers.

Given the newly adopted process, we understand that reviewers are no longer able to participate in discussions or update their scores. We sincerely thank all reviewers for their time and effort, your feedback has been invaluable in improving our submission. We would also like to express our special gratitude to the newly assigned Area Chair. We understand that taking over the evaluation process without the opportunity to engage with the original reviewers is an especially challenging task. We appreciate your dedication to maintaining the scientific integrity of the conference.

**To support the newly assigned AC and PC in this process, we provide below a concise Rebuttal Summary**, outlining the main experiments and clarification we have undertaken in response to the main concerns raised by the reviewers.

**1. Reviewer gbyd – Score: 8 (No response to rebuttal yet)**
- **Main Concern:** Concerned that the current limitations of the editor model may hinder the scalability of Vision-Zero, and requested more parameter ablation studies.
- **Response & Changes:** We simulated training Vision-Zero on a dataset with 20% editing failures. The results show that Vision-Zero still significantly enhances the model's reasoning ability, demonstrating its robustness to failed editing samples and its independence from a strong editor. Additionally, we conducted ablation studies on key parameters and components, including the number of players, number of clue rounds, and role-advantage estimation.


**2. Reviewer CCNi – Score: 4 (Stated would consider raising the score after reading other reviewers' discussions)**
- **Main Concern:** Concerned that performance gains on chart tasks are minor, and requested further explanation of the connection between the “Who is the Spy?” task and general QA tasks.
- **Response & Changes:**
  - **Evaluation on New Benchmarks:** We explained that the modest improvements on chart tasks are due to performance saturation of tasks and model behavioral differences between training and evaluation setups. To illustrate the effective performance improvement on chart tasks, we further validated Vision-Zero on more challenging, reasoning-intensive chart understanding tasks. Results show an average accuracy improvement of **3.9%** across four difficult benchmarks, surpassing most chart-specialized models.
  - **Example Explanation:** We also clarified that the "Who is the Spy?" task enhances general QA performance because the skills required to play it are highly aligned with those needed for general multimodal QA. Moreover, the strategic environment of the game encourages meticulous reasoning and logical thinking. We illustrated this with examples of model responses on general QA tasks before and after training with Vision-Zero.


**3. Reviewer K9Zp – Score: 6 (No response to rebuttal yet)**

- **Main Concern:** Requested clarification on the advantages of Vision-Zero compared to directly training on images edited via image editors or programmatic rendering.
- **Response & Changes:** We explained that, compared with directly training on edited images, Vision-Zero introduces a self-play environment that enables continuous learning. Moreover, Vision-Zero aims to enhance the general reasoning ability of VLMs rather than focusing solely on fine-grained visual comparison or perception skills. We further compared Vision-Zero with direct-training baselines such as MiCo under identical experimental settings, showing that Vision-Zero substantially boosts general reasoning ability.


**4. Reviewer B6Um – Score: 4 (No response to rebuttal yet)**

- **Main Concern:** Request to compare Vision-Zero with simpler multi-image contrast baselines such as MiCo, and to report training efficiency.
- **Response & Changes:** We compared Vision-Zero and MiCo under identical training configurations (dataset, initialization, and hyperparameters). Vision-Zero improves average accuracy across six reasoning datasets by **1.3%** over MiCo. We also directly compared sample efficiency between Vision-Zero and GRPO. The results show that Vision-Zero achieves **2.67×** and **6.4×** sample efficiency over standard GRPO on Qwen2.5-VL-7B and InternVL3-8B, respectively.


We believe these revisions provide strong evidence of Vision-Zero’s effectiveness and novelty.

Sincerely,

The Authors

---

### Meta-Review · Area_Chair_twRy · 2026-01-03

**Summary:**

This paper introduces Vision-Zero, improving VLMs through a game called "Who is the spy". The models will play against each other, and they generate their own training data, which gets rid of the need for slow and expensive human labeling. Most reviewers appreciate the design of this framework, but leave concerns on real-world applicability, computational cost, and whether the proposed game mechanics meaningfully translate to standard benchmark gains.

After the rebuttal, I would say most of the questions and demands from the reviewers, such as adding more ablation studies with GRPO, and listing the expense of creating such data, are addressed.  However, there are two minor concerns still remain. 1. The performance on the ***InternVL family is very marginal***, and the rebuttal from the authors is not satisfactory. However, such phenomena also exist in many other papers. 2. The complicated setup is hard to implement in many real cases.

The AC recommends acceptance, provided that the majority of the suggested revisions are integrated into the final version of the paper.

**Reviewer Concerns:**

Reviewers gbyd, and K9Zp worried the system might break if the image editor made mistakes. The authors ran a "stress test" by intentionally breaking 20% of the image edits and showed that the model still learned effectively, proving the system is remarkably resilient to noise.

Reviewer CCNi wasn't sure why playing a social game makes a model better at general Q&A. The authors explained that the game forces the model to look for tiny visual differences and think several steps ahead—skills that are identical to what’s needed for complex reasoning in any domain.

Reviewer B6Um was concerned that these multi-turn games would be too slow to train. The rebuttal showed that while one round takes longer, the quality of the "lessons" the model learns is so high that it actually reaches top performance 2.6x to 6.4x faster than standard methods.

**Reviewer Scores:**

Reviewer gbyd: 8
Reviewer K9Zp: 6
Reviewer CCNi: 4
Reviewer B6Um: 4

According to the discussion, AC believes that the reviewer CCNi could potentially like to increase the rating.

---

### Decision · Program_Chairs · 2026-01-26

Accept (Poster)